



# Automated Monte Carlo-based Quantification and Updating of Geological Uncertainty with Borehole Data (AutoBEL v1.0)

Zhen Yin[1], Sebastien Strebelle[2], Jef Caers[1]

[1]Department of Geological Sciences, Stanford University, Stanford, CA 94305, USA

[2]Chevron, Houston, USA

*Correspondence to*: Zhen Yin (yinzhen@stanford.edu)

Journal: *Geoscientific Model Development*.

**Abstract.** We provide an automated method for uncertainty quantification and updating of geological models using borehole data for subsurface developments (groundwater, geothermal, oil & gas, and CO2 sequestration, etc.) within a Bayesian

framework. Our methodologies are developed with the Bayesian Evidential Learning protocol for uncertainty quantification. Under such framework, newly acquired borehole data directly and jointly update geological models (structure, lithology, petrophysics and fluids), globally and spatially, without time-consuming model re-buildings. To address the above, an ensemble of prior geological models is first constructed by Monte Carlo simulation from prior distribution. Once the prior model is tested by means of falsification process, a sequential direct forecasting is designed to perform the joint uncertainty

quantification. The direct forecasting is a data-scientific method that learns from a series of bijective operations to establish "Bayes-linear-Gauss" statistical relationships between model and data variables. Such statistical relationships, once conditioned to actual borehole measurements, allows for fast computation posterior geological models. The proposed framework is completely automated in an opensource project. We demonstrate its application by applying to a generalized synthetic dataset motivated by a gas reservoir from Australia. The posterior results show significant uncertainty reduction in

both spatial geological model and gas volume prediction, and cannot be falsified by new borehole observations. Furthermore, our automated framework completes the entire uncertainty quantification process efficiently for such large models.



## 1. Introduction

Uncertainty quantification (UQ) is at the heart of decision making. This is particularly true in subsurface applications such as groundwater, geothermal, fossil fuels, $CO_2$ sequestration, or minerals resources. Uncertainty on the geological structures, rocks and fluids is due to the lack of access to the subsurface geological medium. For most of the subsurface applications, knowledge

of the geological settings is mainly gained through the drilling of well boreholes where geophysical or rock physical measurements are made. For example, from several to tens or hundreds of boreholes are drilled in geothermal or groundwater appraisals (e.g. Le Borgne et al., 2006; Klepikova et al., 2011; Vogt et al., 2010), while in mineral resources and shale gas, the number of boreholes can be up to even thousands (e.g. Curtis, 2002; Territory et al., 2013). From borehole data, geological models are constructed for appraisal and uncertainty quantification, such as estimating water volumes stored in groundwater

systems or heat storage in a geothermal system. Realistic geological modelling involves complex procedures (Caumon, 2010, 2018; de la Varga et al., 2019). This is due to the hierarchical nature of geological formations: fluids are contained in a porous medium, the porous medium is defined by various lithologies, lithological variation is contained in faults and layers (structure). In addition, boreholes are not drilled all at once, but throughout the lifetime of managing the Earth resource.

Representing the unknown subsurface geological reality by a single deterministic model has been commonly used (Beven, 1993; Royse, 2010), mostly by means of a single realization of the structure (layers/faults), rock and fluid model derived from the borehole data with other supporting geological and geophysical interpretations (e.g., Fischer et al., 2015; Kaufmann and Martin, 2008). However, relying on a single model cannot reflect the inherent geological uncertainty (Neuman, 2003). Recent advances in geostatistics have shown the importance of using multiple model realizations for uncertainty quantification in

many geoscience fields, including glaciology (e.g., Cullen et al., 2017), hydrogeology (e.g., Barfod et al., 2018; Zhou et al., 2014), hydrology (e.g. Goovaerts, 2000; Marko et al., 2014), hydrocarbon reservoir modelling (e.g. Caers and Zhang, 2004; Christie et al., 2002; Dutta et al., 2019), geothermal (e.g. Rühaak et al., 2015; Vogt et al., 2010). Geostatistical approaches can provide multiple geological models that are conditioned/constrained to borehole data). When new boreholes are drilled, uncertainty needs to be updated. While uncertainty updating in forms of data assimilation are commonly applied in various

subsurface applications, they are rarely used for updating to newly drilled borehole data, often termed "hard data" in geostatistical literatures (Goovaerts, 1997). Elfeki and Dekking (2007) used coupled Markov chain (CMC) approach to calibrate hydrogeological lithology model by conditioning on boreholes in the central Rhine-Meuse delta from the Netherlands, and then ran Monte Carlo simulation to re-evaluate the hydrogeological uncertainty. Similar approach was also used by Li et al. (2016) to reduce the uncertainty in near-surface geology for the risk assessment of soil slope stability and safety in Western

Australia. Jiménez et al. (2016) updated 3D hydrogeological models by adding new geological features identified from borehole tracer tests. Eidsvik and Ellefmo (2013) and Soltani-Mohammadi et al. (2016) investigated the value of information of additional boreholes for uncertainty reduction in mineral resource evaluations.





The problem of geological uncertainty, due to its interpretative nature and the presence of prior information, is often handled in a Bayesian framework (Scheidt et al., 2018). The key part often lies in the joint quantification of the prior uncertainty on all modeling parameters, whether structural, lithological, petrophysical and fluid. A common problem is that the observed data may lie outside the defined prior model, hence is falsified. Another major issue is that most of the state-of-the-art uncertainty

updating practices deal with each geological model component separately (a silo treatment of each UQ problem). However, the borehole data informs all components jointly, and hence any separate treatment ignores the likely dependency between the model components, possibly returning unrealistic uncertainty quantification. A final concern, more practically, lies around the automating any uncertainty updating. Geological modeling often requires significant individual/group expertise and manual intervention to make the model adhere to geological rules, hence requiring often months of work when new data is acquired.

There is to date, no method that addresses, with borehole data, the falsification, the joint uncertainty quantification and the automation problem.

Recently, a uncertainty quantification protocol termed Bayesian Evidential Learning has been proposed to address decision making under uncertainty, and applied to cases in oil/gas, groundwater contaminant remediation and geothermal energy

(Hermans et al., 2018, 2019; Scheidt et al., 2018). It provides explicit standards that need to be reached at each stage of its UQ design with the purpose of decision making, including, model falsification, global sensitivity analysis, prior elicitation and data-science driven uncertainty reduction under the principle of Bayesianism. The model falsification and data-scientific approaches for uncertainty reduction are what is of concern in this paper. Also, we will deal with one specific data source: borehole data, through logging or coring, for geological model uncertainty quantification. We will address the model

falsification problem involving this data source. We will also extend a method termed direct forecasting (Hermans et al., 2016; Satija et al., 2017; Satija and Caers, 2015) to update uncertainty of all geological model parameters, jointly, using the borehole data. To achieve this, we will present a model formulation that involves sequential updating based on the hierarchy typically found in subsurface formation: structures, then lithology, then property and finally fluid distribution. With a generalized synthetic field case study of uncertainty quantification of gas volume in an offshore reservoir, we will illustrate our approach

and emphasize the need for automation, minimizing the need for tuning parameters that require human interpretation.

## 2. Methodology

### 2.1 Bayesian Evidential Learning

#### 2.1.1 Overview

We establish the geological uncertainty quantification framework based on Bayesian Evidential Learning (BEL), which is

briefly reviewed in this section. BEL is not a method, but a prescriptive & normative data-scientific protocol for designing uncertainty quantification within the context of decision making (Athens and Caers, 2019; Hermans et al., 2018; Scheidt et al., 2018). It integrates four constituents in UQ – data, model, prediction and decision under scientific methods and philosophy of



Bayesianism. In BEL, the data is used as evidence to infer model or/and prediction hypotheses via statistical learnings from the prior distribution, whereas decision making is ultimately informed by the model and prediction hypotheses. The BEL protocol consists of six UQ steps: 1) formulating the decision questions and prediction variables; 2) statement of model parametrization and prior uncertainty; 3) Monte Carlo and prior model falsification with data; 4) global sensitivity analysis

between data and prediction variables; 5) uncertainty reduction based on data-scientific methods that reflect the principle of Bayesian philosophy; 6) posterior falsification and decision making. Bayesian methods, particularly in the Earth Science rely on the statement of prior uncertainty. However, such statement may be inconsistent with data in the sense that the prior cannot predict the data, hence the important falsification step. We provide next important elements of BEL within the problem of this paper: prior model definition, falsification & inversion by direct forecasting.

**2.1.2 Hierarchical model definition**

In geological uncertainty quantification any prior uncertainty statement needs to involve all model components jointly. A geological model $\mathbf{m}$ typically consists of four components that are modelled in hierarchical order: structural model $\chi$ (e.g. faults, stratigraphic horizons), rock types $\zeta$ (which are categorical, e.g. sedimentary or architectural facies), petrophysics model $\kappa$ (e.g. density, porosity, permeability), and subsurface fluid distribution $\tau$ (e.g. water saturation, salinity).

$$\mathbf{m} = \{\chi, \zeta, \kappa, \tau\} \tag{1}$$

The uncertainty model then becomes the following sequential decomposition:

$$f(\mathbf{m}) = f(\chi, \zeta, \kappa, \tau) = f(\tau|\chi, \zeta, \kappa)f(\kappa|\chi, \zeta)f(\zeta|\chi)f(\chi) \tag{2}$$

In addition, because of the spatial context of all geological formations, we divide the model variables into global and spatial

ones. The global variables, such as proportions, depositional system interpretation, or trend, are scalars and not attached to any specific grid locations, whereas the spatial variables are gridded. Here, we term the global variables as $\mathbf{m}_{gl}$, and the spatial as $\mathbf{m}_{sp}$. In this way, the geological model variables are:

$$\mathbf{m} = \left\{ \left(\mathbf{m}_{gl_{\chi}}, \mathbf{m}_{sp_{\chi}}\right), \left(\mathbf{m}_{gl_{\zeta}}, \mathbf{m}_{sp_{\zeta}}\right), \left(\mathbf{m}_{gl_{\kappa}}, \mathbf{m}_{sp_{\kappa}}\right), \left(\mathbf{m}_{gl_{\tau}}, \mathbf{m}_{sp_{\tau}}\right) \right\} \tag{3}$$

The prior uncertainty $f(\mathbf{m})$ of the global and spatial variables needs to be specified for each model component; this is problem

specific and may require substantial amount of work by considering the existing data (e.g. the system is deltaic) and any prior knowledge about the interpreted systems. Using the prior distribution $f(\mathbf{m})$, we run Monte Carlo to generate a set of L model realizations $\left\{\mathbf{m}^{(1)}, \mathbf{m}^{(2)}, \dots, \mathbf{m}^{(L)}\right\}$. This means instantiating all geological variables $\chi, \zeta, \kappa, \tau$ jointly.

Since borehole data provide information at the locations of drilling, we define the data variables $\mathbf{d}$ through an operator $\mathbf{G}_d$.

$$\mathbf{d} = \mathbf{G}_d\mathbf{m} \tag{4}$$

$\mathbf{G}_d$ is simply a matrix in which each element is either 0 or 1 identifying the locations of boreholes in the model $\mathbf{m}$. By applying $\mathbf{G}_d$ to prior geological model realizations, we obtained a set of L samples of the borehole data variable.



$$\mathbf{d} = \left\{\mathbf{d}^{(1)}, \mathbf{d}^{(2)}, \dots, \mathbf{d}^{(L)}\right\} \tag{5}$$

Note that we term the actual acquired data as $\mathbf{d}_{\text{obs}}$.

The prediction variable $\mathbf{h}$, such as storage volume of a ground water aquifer, or the heat storage of a geothermal reservoir, is defined through another operator (linear or nonlinear):

$$\mathbf{h} = \mathbf{G}_{\text{h}}(\mathbf{m}) \tag{6}$$

Applying this function to the prior model realizations we get

$$\mathbf{h} = \left\{\mathbf{h}^{(1)}, \mathbf{h}^{(2)}, \dots, \mathbf{h}^{(L)}\right\} \tag{7}$$

A common problem in practice is that the statement of prior may be too narrow (overconfidence) and hence may not in fact predict the observed data. In falsification, we use hypothetic-deductive reasoning to attempt to reject the prior by means of

data, namely we state the null-hypothesis: the prior can predict the observation and attempt to reject it. This step does not involve matching models to data, it is only a statistical test. One way of achieving this is using outlier detection as discussed in the next section.

### 2.1.3 Falsification using multivariate outlier detection

Our reasoning is that a prior model is falsified if the observed data $\mathbf{d}_{\text{obs}}$ is not within the same population as the samples

$\mathbf{d}^{(1)}, \mathbf{d}^{(2)}, \dots, \mathbf{d}^{(L)}$, i.e. $\mathbf{d}_{\text{obs}}$ is an outlier. Evidently, the data variable can be high-dimensional due to large number of wells with various types of measurements on structure, facies, petrophysics and saturation, which calls for multi-variate outlier detection. We propose in this paper to use a robust statistical procedure based on Mahalanobis distance to perform the outlier detection. The robust Mahalanobis distance (RMD) for each data variable realization $\mathbf{d}^{(\ell)}$ or $\mathbf{d}_{\text{obs}}$ is calculated as:

$$\text{RMD}(\mathbf{d}^{(\ell)}) = \sqrt{(\mathbf{d}^{(\ell)} - \boldsymbol{\mu})^{\text{T}} \boldsymbol{\Sigma}^{-1}(\mathbf{d}^{(\ell)} - \boldsymbol{\mu})} \quad , \text{ for } \ell = 1, 2, \dots, L \tag{8}$$

where $\boldsymbol{\mu}$ and $\boldsymbol{\Sigma}$ are the robust estimation of mean and covariance of the data (Hubert and Debruyne, 2010; Rousseeuw and Driessen, 1999). Assuming $\mathbf{d}$ distributes as multivariate Gaussian, the distribution of $[\text{RMD}(\mathbf{d}^{(\ell)})]^2$ will be Chi-Squared $\chi_{\text{d}}^2$. We will use choose the 97.5 percentile of $\sqrt{\chi_{\text{d}}^2}$ as the tolerance for the multivariate dimensional points $\mathbf{d}^{(\ell)}$. If the $\text{RMD}(\mathbf{d}_{\text{obs}})$ falls outside the tolerance ($\text{RMD}(\mathbf{d}_{\text{obs}}) > \sqrt{\chi_{\mathbf{d},97.5}^2}$), the $\mathbf{d}_{\text{obs}}$ will be considered as outliers, which means the prior model has very small probability to predict the actual observations, hence is falsified. Outlier detection using the Mahalanobis distance

has the advantages of providing robust statistical calculations. In addition, diagnostic plots can be used to visualize the result for high-dimensional data. However, it requires the marginal distribution of data to be Gaussian. If the data variables are not Gaussian, other outlier detection approaches such as one-class SVM (Schölkopf et al., 2001) or Isolation Forest (Liu et al., 2008) can be used.





## 2.2 Direct forecasting

### 2.2.1 Review

If the prior model cannot be falsified, we will use direct forecasting to reduce geological model uncertainty. Direct forecasting (DF) is a prediction-focused data science approach for inverse modeling (Hermans et al., 2016; Satija et al., 2017; Satija and

Caers, 2015). The aim is to estimate/learn the conditional distribution $f(\mathbf{h}|\mathbf{d})$ between the prediction variable $\mathbf{h}$ and data variable $\mathbf{d}$ from prior Monte Carlo samples. Then, instead of using traditional inverse methods that require re-building models to update prediction, direct forecasting directly calculates the conditional prediction distribution $f(\mathbf{h}|\mathbf{d}_{obs})$ through the statistical learning based on data. The learning strategy of direct forecasting is that, by employing bijective operations, the non-Gaussian problem $f(\mathbf{h}|\mathbf{d})$ can be transformed into a linear-Gauss problem of transformed variables $(\mathbf{h}^*, \mathbf{d}^*)$:

$$\mathbf{h}^* \sim \exp\left(-\frac{1}{2}\left(\mathbf{h}^* - \mathbf{h}^*_{prior}\right)^T C^{-1}_{prior}\left(\mathbf{h}^* - \mathbf{h}^*_{prior}\right)\right); \mathbf{d}^*_{obs}; \mathbf{d}^* = G\mathbf{h}^* \qquad (9)$$

This makes $f(\mathbf{h}^*|\mathbf{d}^*_{obs})$ become a "Bayes-linear-Gauss" problem that has an analytical solution:

$$E[\mathbf{h}^*|\mathbf{d}^*_{obs}] = \mathbf{h}^*_{posterior} = \mathbf{h}^*_{prior} + C_{prior}G^T\left(GC_{proir}G^T\right)^{-1}\left(\mathbf{d}^*_{obs} - G\mathbf{h}^*_{prior}\right)$$

$$Var[\mathbf{h}^*|\mathbf{d}^*_{obs}] = C_{posterior} = C_{prior} - C_{prior}G^T\left(GC_{proir}G^T\right)^{-1}GC_{prior} \qquad (10)$$

In detail, the specific steps of direct forecasting are:

1. Monte Carlo: generate L samples of prior model, and run forward function to evaluate data and prediction variables.

  2. Orthogonality: PCA (Principal Component Analysis) on data variable $\mathbf{d}$ and prediction variable $\mathbf{h}$.

  3. Linearization: maximize linear correlation between the orthogonalized data and variables by Normal Score Transform and CCA (Canonical Component Analysis), obtaining transformed $\mathbf{h}^*, \mathbf{d}^*$.

  4. Bayes-linear-Gauss: calculate conditional mean and covariance of the transformed prediction variable

5. Sampling: sample from the posterior distribution of transformed prediction variable $\mathbf{h}^*_{posterior}$

  6. Reconstruction: invert all bijective operations, obtaining $\mathbf{h}_{posterior}$ in the original space.

One key question in direct forecasting is how to determine the Monte Carlo samples size L. Usually, the samples size L lies between 100-1000, according to its successful applications in water resources (Satija and Caers, 2015), hydrogeophysics (Hermans et al., 2016), hydrocarbon reservoirs (Satija et al., 2017). This is also true because, for most cases, the prediction

variables are much simpler than model variables.

The direct forecasting can also be extended to update model variables, by simply replacing the prediction variable $\mathbf{h}$ by model variable $\mathbf{m}$ in the above algorithms, to obtain $f(\mathbf{m}|\mathbf{d}_{obs})$ without conventional model inversions (Park and Caers, 2019). However, the high dimensionality of spatial models (millions of grid cells) imposes challenge to such extension. This is

because CCA requires that the sum of input data and model variables dimensions to be smaller than the Monte Carlo samples size L: $L > dim(\mathbf{d}) + dim(\mathbf{m})$. Otherwise it will always produce perfect correlations (correlation coefficients be 1) (Pezeshki et al., 2004). Although PCA can significantly reduce the dimensionality of $\mathbf{m}$ from L×P to L×L, where P is the number of





model grid cells and L≪P, this requirement is still difficult to meet. Global Sensitivity Analysis is therefore applied, to select a subset of the PCA orthogonalized **m** that is most informed by the data variables. In this paper, we use a Distance-Based Generalized Sensitivity Analysis (DGSA) method (Fenwick et al., 2014; Park et al., 2016) to perform sensitivity analysis. As a regionalized global sensitivity analysis approach, DGSA has its specific advantages for high-dimensional problems while

requiring no functional form between model responses and model parameters. It can efficiently compute global sensitivity, which makes it preferred for our geological UQ problem where the models are large and computationally intensive.

### 2.2.2 Direct forecasting on a sequential model decomposition

We defined our prior uncertainty model (Eq.2) through a sequential decomposition of hierarchical model components. Likewise, the conditioning of such model components to borehole data will be done, using direct forecasting in a sequential

fashion:

$$f(\chi, \zeta, \kappa, \tau | \mathbf{d}_{obs}) = f(\tau | \chi, \kappa, \zeta, \mathbf{d}_{obs,\tau}) f(\kappa | \chi, \zeta, \mathbf{d}_{obs,\kappa}) f(\zeta | \chi, \mathbf{d}_{obs,\zeta}) f(\chi | \mathbf{d}_{obs,\chi}) \tag{11}$$

Following this equation, the joint uncertainty quantification is equivalent to a sequential uncertainty quantification, where

uncertainty quantification of one model component conditions to borehole data and posterior models of the previous components. Basically, the posterior model of $\chi$ becomes prior model for $\zeta$ and so on. Direct forecasting has not been applied within this framework of Eq (11), hence this is one of the new contributions in this paper. In applying direct forecasting we will use the posterior realizations of $\chi$ and prior realizations of $\zeta$ to determine a conditional distribution $f(\zeta | \chi_{posterior})$, then we evaluate this using borehole observations $\mathbf{d}_{obs,\zeta}$ of $\zeta$.

To apply this framework to discrete variables such as lithology, we need a different method for dimension reduction than using PCA. PCA relies on a reconstruction by linear combination of principal component vectors, which becomes challenging when the target variable is discrete (see Figure 1). To avoid this, a level set method of signed distance function (Osher and Fedkiw, 2003; Deutsch and Wilde, 2013) is employed to transform rock type models into a continuous scalar field of signed distances

before applying PCA. Here, considering S discrete rock types in model $\zeta$, for each s-th (s = 1, 2, ..., S) rock type, the signed distance $\psi(\mathbf{x})$ from location $\mathbf{x}$ to its closest different rock type $\mathbf{x}_\beta$ can be computed as:

$$\psi(\mathbf{x}) = \begin{cases} +\|\mathbf{x} - \mathbf{x}_\beta\|, & \text{if } \zeta(\mathbf{x}) = s \\ -\|\mathbf{x} - \mathbf{x}_\beta\|, & \text{otherwise} \end{cases} \qquad s = 1, 2, ..., S \tag{12}$$

Figure 2 illustrates the concept of using a signed distance function to first transform a sedimentary lithology model to continuous signed distances for PCA. We observe that, with the signed distance as an intermediate transformation, the inverse

PCA recovers the lithology model. In the case of multiple categories, we will have multiple signed distance functions.





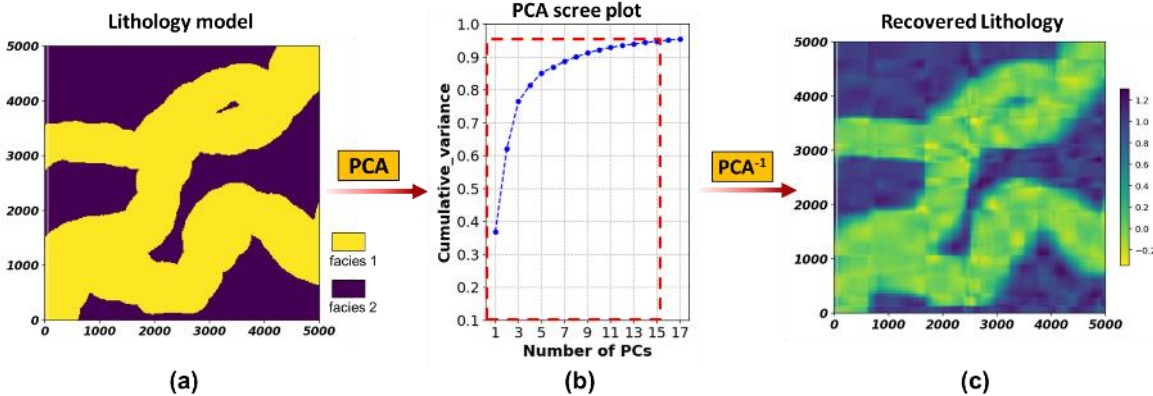

**Figure 1. PCA on discrete lithology model: (a) the original lithology model (b) Scree plot of PCA on the lithology model. (c) The reconstructed model from inverse PCA using the preserved PCs (marked by the red dash line on the scree plot).**

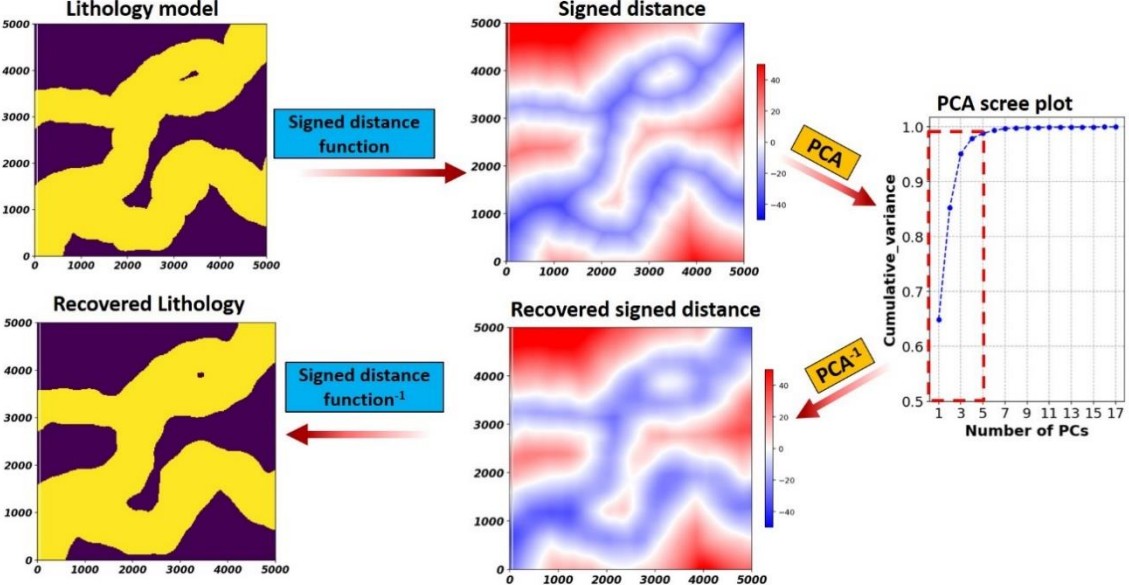

5    **Figure 2: Example of transforming categorical lithology model to continuous signed distances for performing PCA.**

### 2.3 Automation and Code

Our objective of automation is to allow for seamless uncertainty quantification once the prior uncertainty models has been established. Therefore, following the above described geological UQ strategies, we design a workflow in Figure 3 to automate the implementation. The workflow starts with the prior model Monte Carlo (MC) samples and borehole observations as input.

10    All following steps including extraction of borehole data variables, prior falsification, sequential direct forecasting, posterior prediction and falsification (if required) are completely automated. With this workflow, we develop an open source Python implementation to execute the automation (named "Auto-BEL"). This opensource project can be accessed from Github (repository: https://github.com/sdyinzhen/AutoBEL-v1.0, DOI: 10.5281/zenodo.3479997). Figure 4 briefly explains the





structure of the Python implementation. This automation implementation allows that, once new borehole observation and the current model state (the prior model) is provided from "*Input*" directory, the uncertainty quantitation and updating can be performed automatically by simply running the Jupyter Notebook "*Control panel*". The results from the automated uncertainty quantification are stored in the "*Output*", classified as "*Model*", "*Data*", and "*Prediction*".

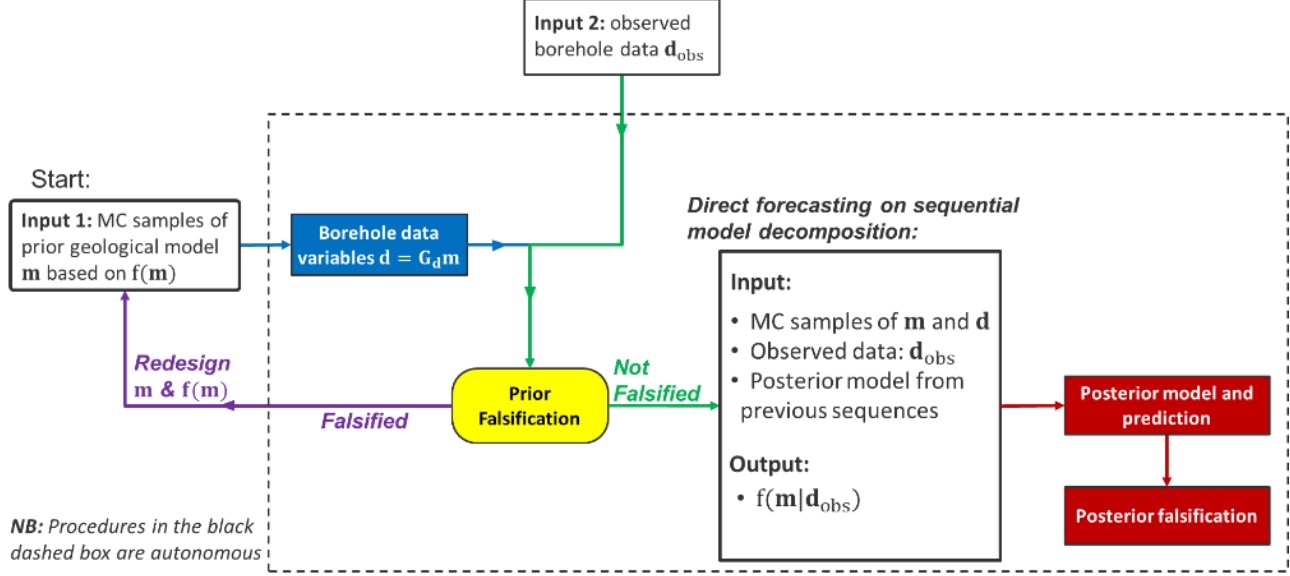

**Figure 3. Proposed workflow to automate the geological uncertainty quantification.**

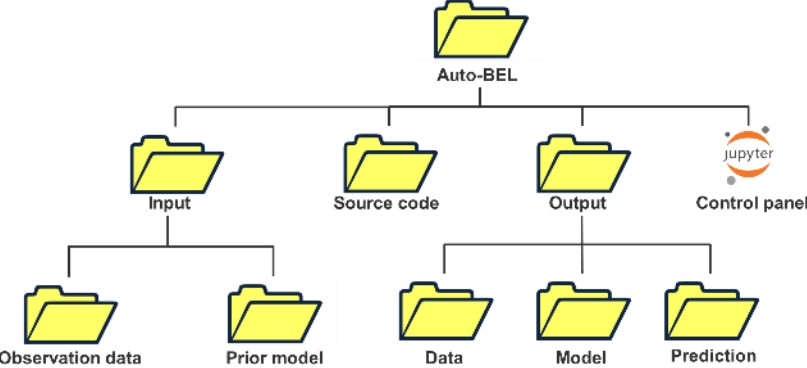

**Figure 4. The structure of the Auto-BEL python implementation project.**

**3. Application example**

**3.1 The field case**

We demonstrate the application of the automated UQ framework using synthetic dataset inspired by a gas reservoir located offshore Australia. Its spatial size is around 50 km (EW) ×25km (NS) with thickness ranging from 75 meter to 5 meters. The reservoir deposits at shallow marine environment, with four lithological facies belts corresponding to four different types of





porous rocks (Figure 5a). The rock porous system contains natural gas and formation water. The major challenges lies in quantifying geological uncertainty due to spatial heterogeneity, appraise gas storage reserve (namely gas initially in place, or GIIP), and then fast update the uncertainty quantification when new boreholes are drilled. This will directly impact the economic decision making for reservoir development.

Initially, the reservoir geological variation is represented on a 3D model (Figure 5b) with a total of 1.5million grid cells with dimension of 200 ×100 ×75 (layers). Companies often drill exploration and appraisal wells before going ahead with producing the reservoir. They would like to decrease uncertainty by such drilling to a point where the risk is considered tolerable to start actual production. To mimic such setting, we consider that initially 4 well-bores (w1, w2, w3, w4, marked in Figure 5b) have

been acquired and that models have been built using the data from these wells. Then 9 new wells (w5 to w13 in Figure 5b) are drilled, and uncertainty needs to be updated. The idea is to use the 9 new wells to automatically update the reservoir uncertainty using the above developed procedures. In order to validate our results, we will use observations from w7 to w13 to reduce the uncertainty, whereas observations from w5 and w6 to analyze the obtained uncertainty quantification.

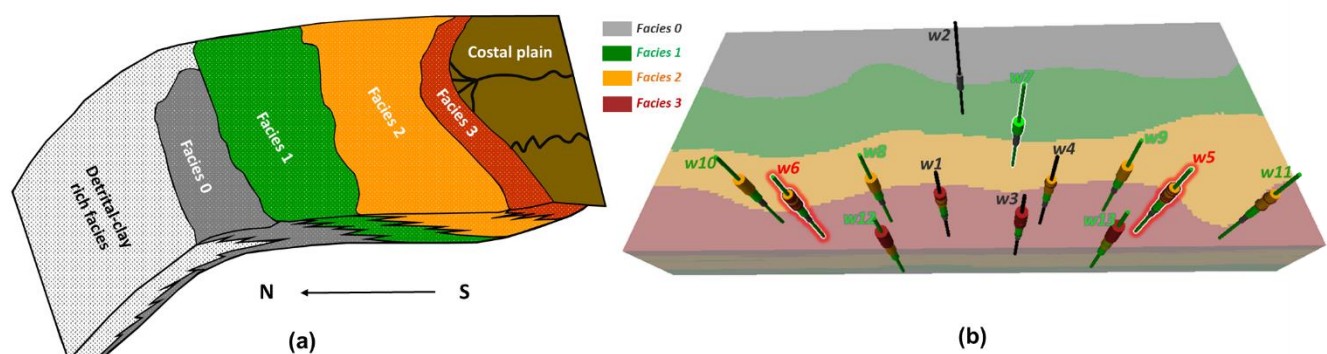

**Figure 5. (a) The field geology conceptual model with the four facies belts. (b) The initial 3D geological model of facies with locations of existing boreholes and newly drilled boreholes.**

### 3.2 Prior model parameterization and uncertainty

### 3.2.1 Approaches

The reservoir geological properties responsible for reserve appraisals are spatial variations of (1) reservoir thickness, spatial
distributions of (2) lithological facies belts, (3) 3D porosity, (4) 3D formation water (saturation); while the spatial heterogeneity of (5) 3D permeability is critical to future production of gas, but not used in volume appraisal. Constructing a prior uncertainty model for these properties requires a balance between considering aspects of the data and overall interpretation based on such data. The strategy in the BEL framework is not to state too narrow uncertainty initially, rather to explore a wide range of possibilities. Based on interpretation from data, Table 1 containing all uncertainties and their prior distribution was constructed.
We will clarify how these uncertainties were obtained.





**Table 1. The global model parameter $m_{gl}$ and its prior uncertainty distribution $f(m_{gl})$. The initial prior distributions of the parameters are mostly assumed to be uniform (formulated as U[min, max]) due to limited available data.**

| Model | Global parameters: $m_{gl}$ | Prior uncertainty: $f(m_{gl})$ | Source for prior uncertainty statement |
|---|---|---|---|
| Reservoir Thickness | Thickness expectation – $Z_{mean}$ | U[36, 51] meter | Seismic interpretations, initial borehole measurements. |
| | Variogram range of trend – $T_{range}$ | U[10000, 40000] meter | |
| | Variogram sill of trend – $T_{sill}$ | U[350, 650] | |
| | Variogram range of residual – $R_{range}$ | U[1000, 5000] meter | |
| | Variogram sill of residual – $R_{sill}$ | U[4, 100] | |
| Lithological Facies | Proportion of facies 1 – fac1 | U[0.22, 0.36] | Boreholes gammy ray logs, seismic amplitude maps, |
| | Proportion of facies 2 – fac2 | U[0.07, 0.27] | |
| | Proportion of facies 3 – fac3 | U[0.13, 0.19] | |
| Porosity & Permeability | Porosity mean in facies 1 – $\phi$1 | U[0.175, 0.225] | Borehole neutron porosity logs, laboratory measurements on core samples |
| | Porosity mean in facies 2 – $\phi$2 | U[0.275, 0.325] | |
| | Porosity mean in facies 3 – $\phi$3 | U[0.225, 0.275] | |
| | Porosity mean in facies 0 – $\phi$0 | U[0.125, 0.175] | |
| | Variogram range of porosity – $\phi_{range}$ | U[4000,10000] meter | |
| | Variogram sill of porosity – $\phi_{sill}$ | U[0.0015 0.003] | |
| | Correlation coeff. between Porosity and log-perm – $r_{\phi k}$ | Normal(0.85, 0.0025) | |
| | log-perm mean in facies 1 – k1 | U[0.3, 1.3] log(mD) | |
| | log-perm mean in facies 2 – k2 | U[1.6, 2.6] log(mD) | |
| | log-perm mean in facies 3 – k3 | U[1, 2] log(mD) | |
| | log-perm mean in facies 0 – k0 | U[-1.6, -0.6] log(mD) | |
| | Variogram range of permeability – $k_{range}$ | U[4000,10000] meter | |
| | Variogram sill of permeability – $k_{sill}$ | U[0.9, 1.4] | |
| Saturation (Sw) | Coeff. a of Eq.14 (capillary pressure model) – a | U[0.041, 0.049] | Laboratory capillary pressure experiments on rock core and fluid samples |
| | Coeff. b of Eq. 14 – b | U[0.155, 0.217] | |
| | Coeff. c of Eq. 14 – c | U[0.051, 0.203] | |





**Thickness**

First, the thickness uncertainty is mainly due to limited resolution of the geophysical seismic data (not shown in this paper). Seismic interpretations show no faults in the geological system, but the thickness variations follow a structural trend. To model thickness uncertainty, we decompose thickness $Z(\mathbf{x})$ into an uncertain trend $T(\mathbf{x})$ and uncertain residual $R(\mathbf{x})$

$$Z(\mathbf{x}) = T(\mathbf{x}) + R(\mathbf{x}) \tag{13}$$

Note that most common geostatistical approaches do not consider uncertainty in trend. Uncertainty in $T(\mathbf{x})$ is based on the seismic interpretation uncertainty. We describe uncertainty on trend using a 2D Gaussian process (Goovaerts, 1997) with

uncertain expectation and spatial covariance. The expectation is based on the single seismic interpretation and the estimated seismic vertical resolution, which is approximately 15 meters. The uncertain spatial covariance of the trend is modeled using a geostatistical variogram with uncertain range (spatial correlation length) and sill (variance). The uncertainty in range and sill are determined from variogram modeling on the single seismic interpretation. Since seismic does not cover the full spatial variation of thickness, a high-varying residual term is needed to describe realistically its variation. The residual $R(\mathbf{x})$ is

modeled using a zero-mean 2D Gaussian process with unknown spatial covariance. This term is highly uncertain, in particular the covariance, because the residual term is observed only at 4 initial borehole locations. However, the variogram range is assumed to be much smaller than the seismic/trend variogram, as residuals aim to represent more local features. Once the Gaussian process is defined, it can be constrained (conditioned) to the actual thickness observation at the vertical boreholes through the generation of conditional realizations. Note that these conditional realizations contain the uncertainties of trend

and residual terms (Figure 6).

**Facies**

The lithological facies are considered to have rather simple spatial variability and described as "belts" (see Figure 5a). These are common in the stratigraphic progression, typical of shallow marine environments. To describe such variation, we use a 3D

Gaussian process that is truncated (Beucher et al., 1993), thereby generating discrete variables. This truncated Gaussian process has specific advantage in reproducing simple organizations of ordered lithologies, thus making a useful model in our case. Because 4 facies exist, three truncations need to be made on the single Gaussian field. The truncation bounds are determined based on facies proportions. The uncertain facies proportions are obtained from lithological interpretations on borehole gamma ray logs and field seismic amplitude maps.

**Porosity and permeability**

For each facies belt, rock porosity and permeability (logarithmic scale, termed as log-perm) are modelled, using two correlated 3D Gaussian processes. The cross-covariances of these processes are determined via Markov-models (Journel, 1999) that only require the specification of a correlation coefficient. Laboratory measurements on the borehole rock core samples show that





permeability is linearly correlated to porosity with a coefficient 0.85, and a small experimental error (around 6% random error according to the lab scientists by repeating the experiments). The marginal distributions of porosity and log-perm are assumed to be normal but with uncertain mean and variances. The mean of porosity and log-perm is based on borehole neutron porosity logs and core sample measurements. Similar to the thickness residual modelling, the spatial covariances are modeled via a

variogram respectively for porosity and permeability, with uncertain range and sill. Limited wellbore observations make variogram range and sill highly uncertain, and therefore large uncertainty bounds are assigned.

**Saturation**

Rocks contain gas and water; hence the uncertain saturation of water (Sw) will affect the uncertain gas volume calculations.

The modelling of Sw is based on a classical empirical capillary pressure model from Leverett J-function (Leverett et al., 1942), formulated as:

$$Sw = 10^{-a*[\log(j)]^2 - b*\log(j) - c} \tag{14}$$

where $j = 0.0055 * h\sqrt{\emptyset/\mathbf{k}}$ , and h is the reservoir depth. The uncertainty parameters in this fluid modelling are the coefficients a, b, c. Their prior distributions are provided by capillary pressure experiments using rock core plugs and reservoir fluids as shown in Table 1.

### 3.2.2 Monte Carlo

By running Monte Carlo from the given prior distribution in

3.2.2 Monte Carlo

, a set of 250 geological model realizations are generated. Figure 6 displays Monte Carlo realizations of the geological model: thickness trend and corresponding thickness model, facies, porosity, permeability (log-perm) and Sw. With prior samples of geological model, prior prediction of GIIP are calculated, using the following linear equation:

$$GIIP = study\,area \times thickness \times porosity \times (1 - Sw)/Bg \tag{15}$$

where the Bg is the gas formation volume factor provided from laboratory measurements. The calculated GIIP prediction is plotted in Figure 7. The plot shows that the initial prediction of reservoir gas storage volume has wide range, which means significant risk can exist during decision makings for field development.





**Figure 6. Layer view of prior Monte Carlo model samples of thickness trend and corresponding thickness, facies, porosity, permeabilty (logarithmic, termed as log-perm), and Sw.**



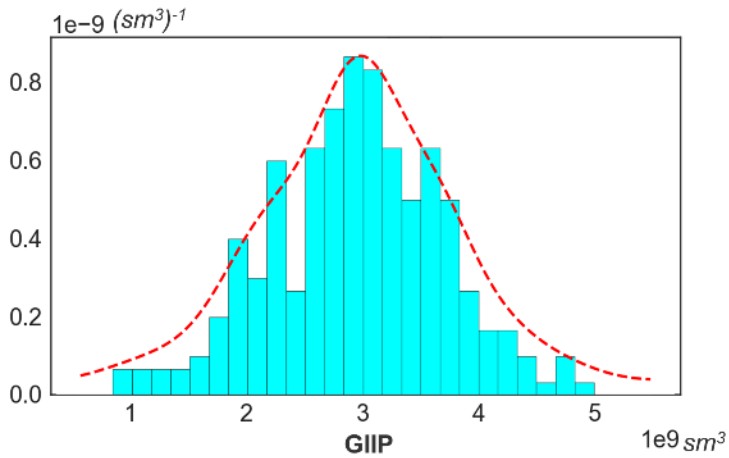

**Figure 7 Uncertainty quantifiation of GIIP based on prior uncertainty and 4 boreholes.**

### 3.3 Prior falsification with newly acquired borehole data

Table 1 is a subjective statement of prior uncertainty. When new data is acquired, this statement can be tested, using a statistical
5  test (section 2.1.3) that may lead to a falsified prior. To perform falsification, borehole data variables at the seven new well
locations (from w7 to w13) are extracted by applying the data forward operator $\mathbf{G}_d$ to the 250 prior model realizations. It
simply means extracting all thickness, facies, petrophysics and saturation at the borehole locations in the prior model. For the
2D thickness model, the new boreholes provide seven data extraction locations. For the 3D model of facies, porosity,
permeability and Sw, each vertical borehole drills through 75 grid layers, thus the seven boreholes provide 2100 extracted data
10  measurements (75 data measurements/well × 7 wells × 4 model components = 2100 data measurements). The dimensionality
of data variable $\mathbf{d}$ in this case therefore equals to 2107. The actual observations of these data ($\mathbf{d}_{obs}$) are measured from the
borehole wireline logs. As described in section 2.1.3, prior falsification is then conducted by applying the Robust Mahalanobis
Distance outlier detection to $\mathbf{d}$ and $\mathbf{d}_{obs}$. Figure 8 shows the calculated RMD for $\mathbf{d}_{obs}$ and the 250 samples of $\mathbf{d}$, where the
distribution of the calculated RMD($\mathbf{d}$) falls to a Chi-Squared distribution, with the RMD($\mathbf{d}_{obs}$) falls below the 97.5 percentile
15  threshold. This shows with (97.5) confidence that the prior model is not wrong.



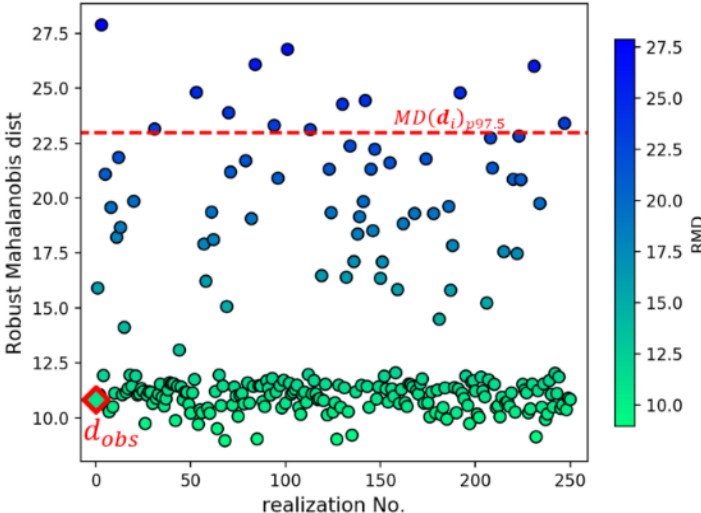

**Figure 8. Prior falsification using robust Mahalanobis Distance (RMD). Circle dots represent the caculated RMD for data variable samples. The red-squared dot is the RMD for borehole observations. The red dash line is the 97.5 percentile of the Chi-Squared distributed RMD.**

## 3.4 Automatic updating of uncertainty with new boreholes

After attempting to falsify the prior uncertainty model, we use the automated framework to jointly update model uncertainty with the new boreholes. The joint model uncertainty reduction is performed sequentially as explained in section 2.2.2. Under the AutoBEL GitHub repository instruction (https://github.com/sdyinzhen/AutoBEL-v1.0/blob/master/README.md), we also provide a supplement YouTube video to demonstrate how this automated update is performed.

### 3.4.1. Thickness and facies

Uncertainty in facies and thickness can be updated jointly, as they are two independent components of the geological model. The Auto-BEL first transforms the categorical facies to continuous model using signed distance function. The transformed signed distances are then combined with thickness model to perform orthogonalization using mixed PCA (Abdi et al., 2013). As shown in Figure 9, the first eigen-image (first principal components (PC1)) of thickness reflects the global variations of reservoir thickness, while higher order eigen-images (e.g. eigen-image of PC40) represent more local variation features. To evaluate what model variables impact thickness variation at the boreholes, DGSA (Fenwick et al., 2014) is then performed to analyze the sensitivity of model variables to data. Figure 10(a) plots the main effects in a Pareto plot. As shown in the plot, DGSA identifies sensitive (measure of sensitivity >1) and non-sensitive (measure of sensitivity <1) model variables. Thickness global parameters of both trend ($Z_{mean}$, $T_{range}$, $T_{sill}$) and residuals ($R_{range}$) show sensitivity to the borehole data. In terms of facies, proportions of the facies 1 (fac1) and 2 (fac2) are sensitive. There are totally 26 sensitive principal components from the spatial model. These sensitive global variables and principal component scores are now selected for uncertainty quantification. Following the steps of direct forecasting (see section 2.2.1), uncertainty reduction proceeds by mapping all





sensitive model variables into a lower dimensional space such that the "Bayes-linear-Gauss" model can be applied. This requires the application of CCA to the selected model variables and data variables, then normal score transformation. Figure 10b shows two examples of cross-plot between model and data variables of the first and tenth canonical components, where we observe linear correlation coefficient of 0.84 even for the tenth canonical components. Once the Bayesian model is

5    specified, one can sample from the posterior distribution and back-transform from lower-dimensional scores into actual facies and thickness models. Figure 10c shows the distribution of the posterior model realizations in comparison to the corresponding prior, showing the reduction of the model uncertainty. Figure 10d shows the comparison between the prior and posterior distributions of the scores for the first 4 sensitive PCs, where the reduction of uncertainty is observed (while noting that uncertainty quantification involves all the sensitive PC score variables). For unselected (non-sensitive) model variables, they

10   remain random according to their prior empirical distribution.

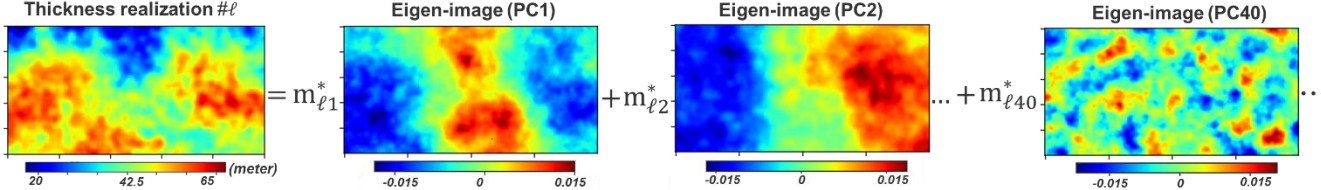

**Figure 9. Example of applying PCA on thickness model. One model realization $\ell$ ($\ell = 1, 2, ..., L$) can be represented by the linear combination of eigen-images scaled by the PC scores $m_\ell^*$.**





**Figure 10. Uncertainty reduction of thickness and facies: (a) global sensitivity of model parameters to borehole data. (b) First and tenth canonical covaraites of data and model variables. The dash redline is the observation data. (c) Posterior and prior distributions of model variables (first and tenth canonical components, corresponding to b). (d) Prior and posterior PC score distributions of first 4 sensitive PCs.**





Figure 11 plots the reconstructed posterior global parameters in comparison to the prior. Uncertainty reduction of sensitive global parameters is observed, while the uncertainty of non-sensitive global parameters ($R_{sill}$ and fac3) is unchanged from their prior. To assess the reconstructed posterior spatial model realizations, the mean and variance are calculated, namely "ensemble mean" and "ensemble variance". Figure 12 and Figure 13 respectively show the ensemble mean and variance of

5    the posterior thickness and facies models. The results in Figure 12 imply that the posterior model thickness is thicker on average than the prior. This change mainly occurs in areas where the new boreholes are drilled. Referring to the actual borehole observations plotted on Figure 12, we also find that the posterior thickness adjusts to the borehole observations at both training (w7-w13) and validating (w5, w6) locations. This improvement is significant compared to the prior model. Furthermore, the ensemble variances (Figure 13) are reduced in the posterior model, mostly in vicinity of the new boreholes. This implies

10    reduction of the spatial uncertainty. One should note that our method does (not yet) result in an exact match of the thickness at borehole data. This is an issue we will comment on in the discussion and conclusion section.

For the facies model, the magnitudes of the uncertainty reduction are not as remarkable, because prior uncertainty at borehole locations was small to start with.

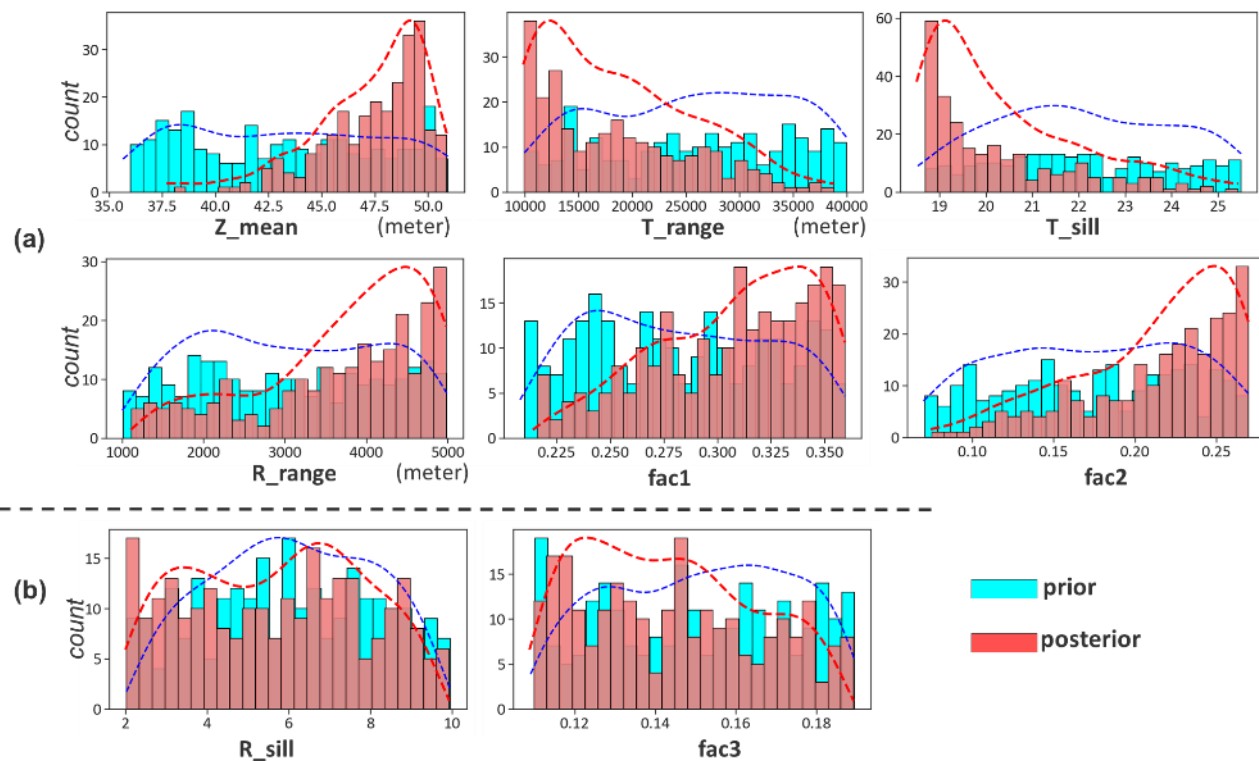

**Figure 11. Uncertainty updating of (a) sensitive, and (b) non-sensitive global model parameters at the first sequence.**


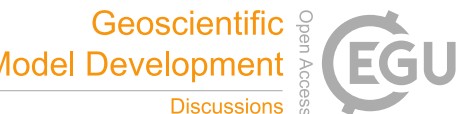

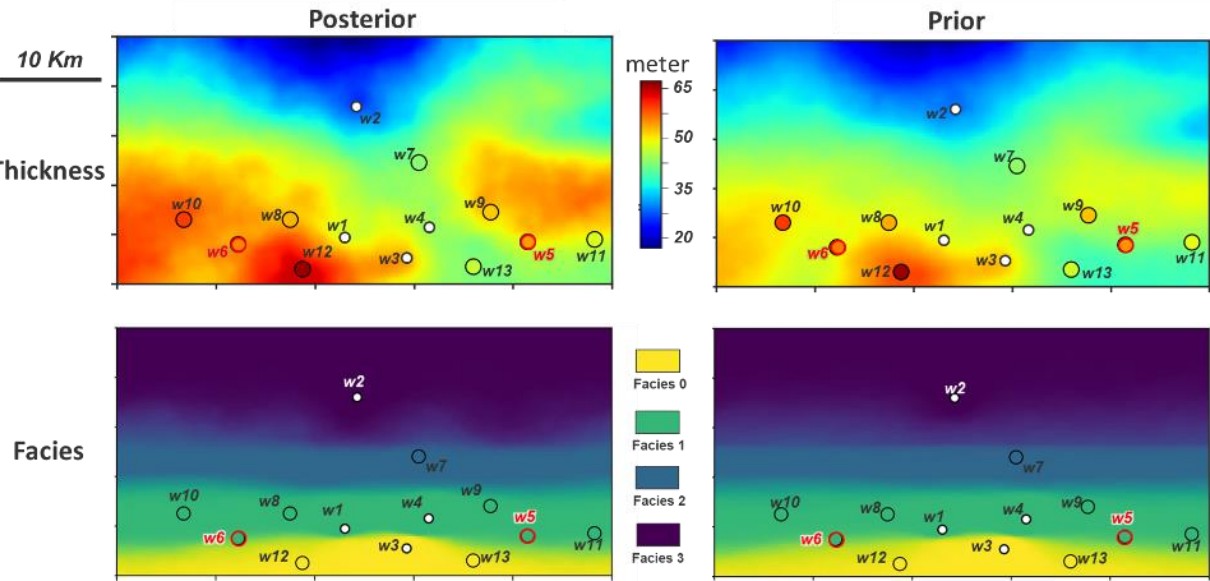

**Figure 12. Ensemble mean of the posterior and prior thickness and facies models. The dots are the borehole locations, where the color represents the actual borehole observation values. The boreholes and models share the same color legend.**

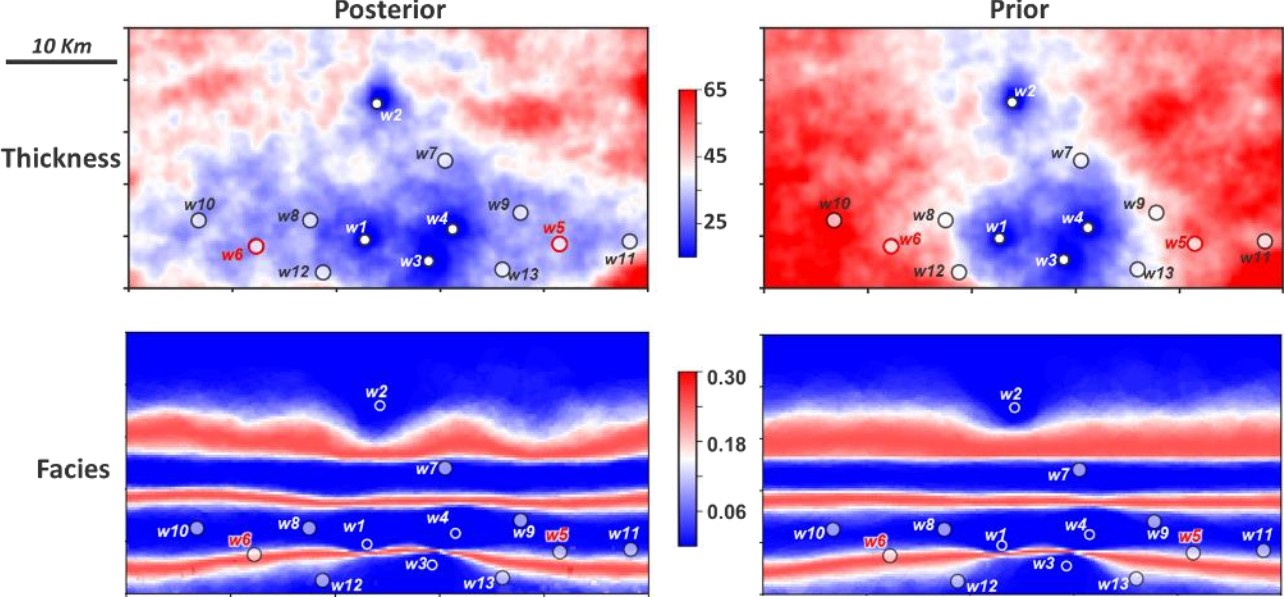

5    **Figure 13. Ensemble variance of the posterior and prior thickness and facies models from the first sequence.**

### 3.4.2 Porosity, permeability and saturation

Auto-BEL is now applied to update uncertainty on porosity, permeability and saturation. This is performed by simply repeating the above automated steps but using posterior models of previous model components as additional constraint. Figure 14, Figure 15 and Figure 16 show the results. In Figure 14, we see sensitive global and spatial model variables that are selected for



uncertainty reduction. Figure 15 shows the constructed the linear correlation between data and sensitive model variables by means of CCA. Figure 16 plots the posterior model realizations (250 realizations) computed from the "Bayes-linear-Gauss" model, where reduced uncertainty is observed when comparing to the prior. The posterior spatial model PC scores are also plotted in Figure 17.

Finally, by back-transformation, we can reconstruct all original model variables. Figure 18 compares ensemble means and variances of the reconstructed posterior porosity, log-perm and Sw, to their corresponding prior models, with actual borehole observations plotted on the top. Taking w7 for example, the actual borehole observations show low values of porosity, permeability and Sw, while the prior model initially expects those values to be large at this location. This is adjusted in the posterior. From the ensemble variance maps, we notice that spatial uncertainty is significantly reduced from prior to posterior in areas near w7. The updates of model expectations and reduction of spatial uncertainty are also observed from the other wells. It implies that the posterior models have been constrained by the borehole observations.

Figure 19 shows one example realization of the spatial models. It shows that, same as the hierarchical order in the prior (Figure 19a), the spatial distributions of posterior porosity and log-perm follow the spatial patterns of their corresponding facies belts (Figure 19b). However, if the joint model uncertainty reduction is performed without the sequential decomposition (not using the posterior models from previous components as additional constraints in direct forecasting), the model hierarchy from facies to porosity and permeability is lost (marked by the purple boxes on Figure 19c). This is because they are treated as independent model variables, which violates the imposed geological order of variables. The linear correlation between porosity and log-perm is also preserved due to the sequential decomposition. We observe similar correlation pattern and coefficient from prior (Figure 20a) to posterior (Figure 20b). But without sequential decomposition, this important feature cannot be maintained as the results shown from Figure 20c: 1) the four clouds pattern (representing the four facies) of the covariate distribution between porosity and log-perm is lost; 2) the correlation coefficient has changed significantly.



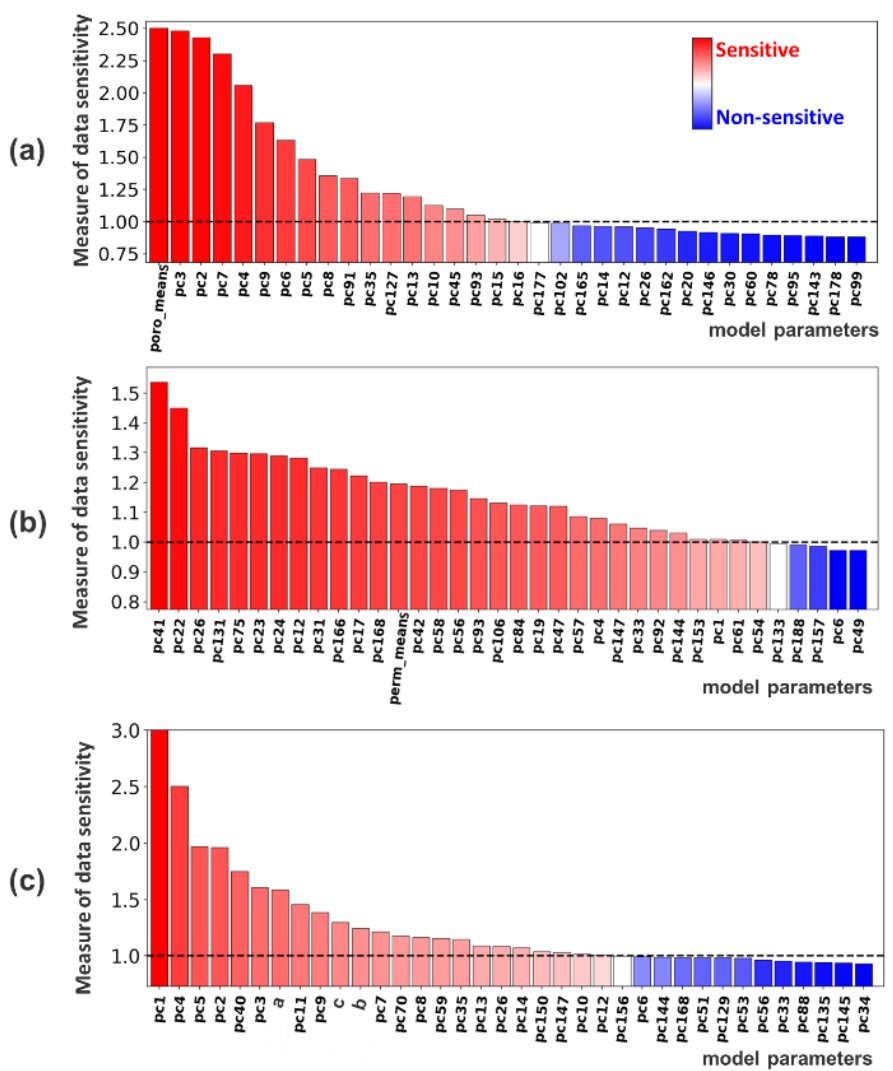

**Figure 14. Results from global sensitivity analysis using DGSA at (a) porosity (b) log-perm and (c) Sw.**

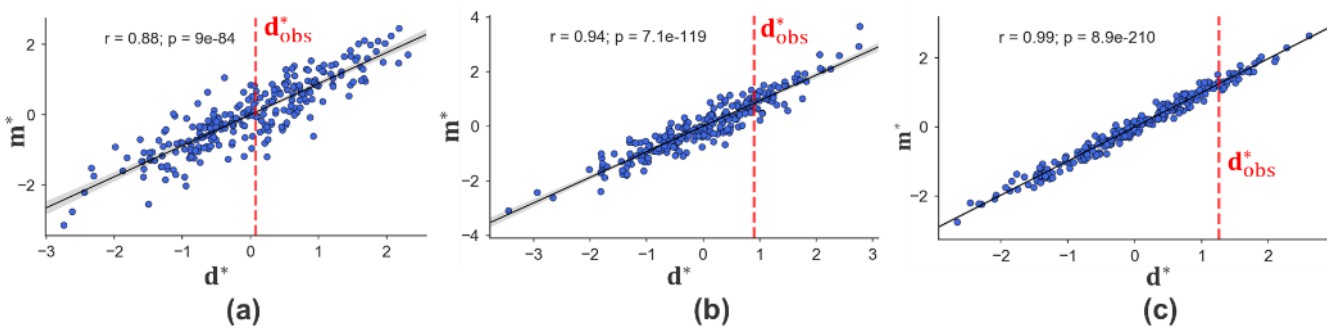

**Figure 15 First canonical covaraites of data and model variables from (a) porosity (b) log-perm and (c) Sw.**



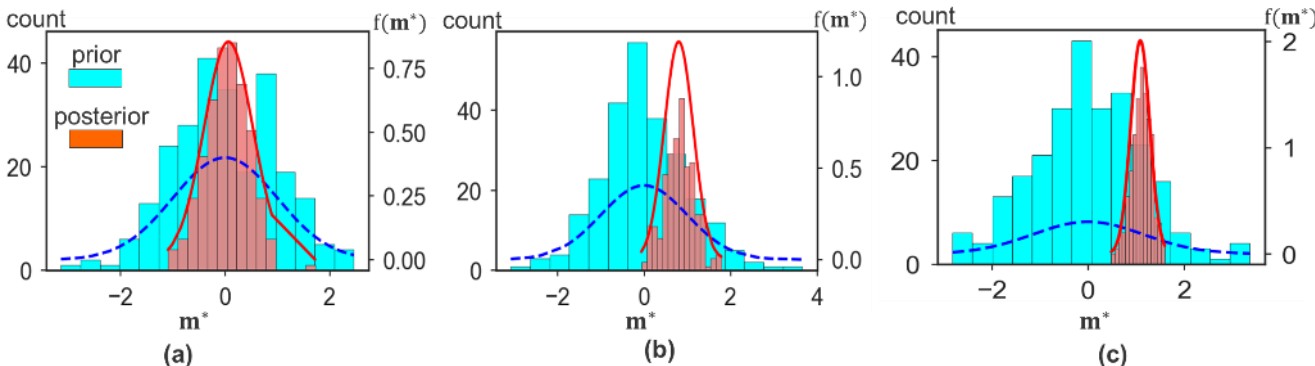

**Figure 16. Reduction of uncertainty of the first model canonical component: (a) porosity (b) log-perm and (c) Sw**

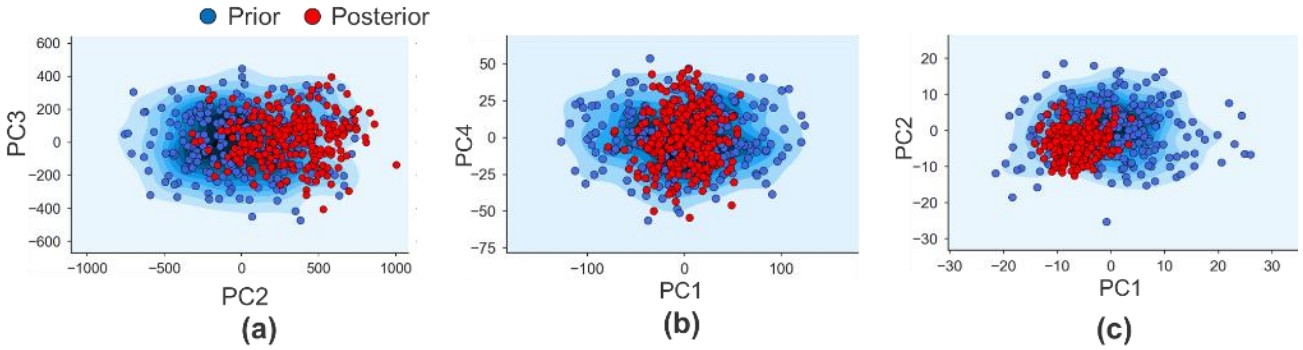

**Figure 17. Prior and posterior distribution of the scores of the first two sensitive PCs: (a) porosity (b) log-perm and (c) Sw.**

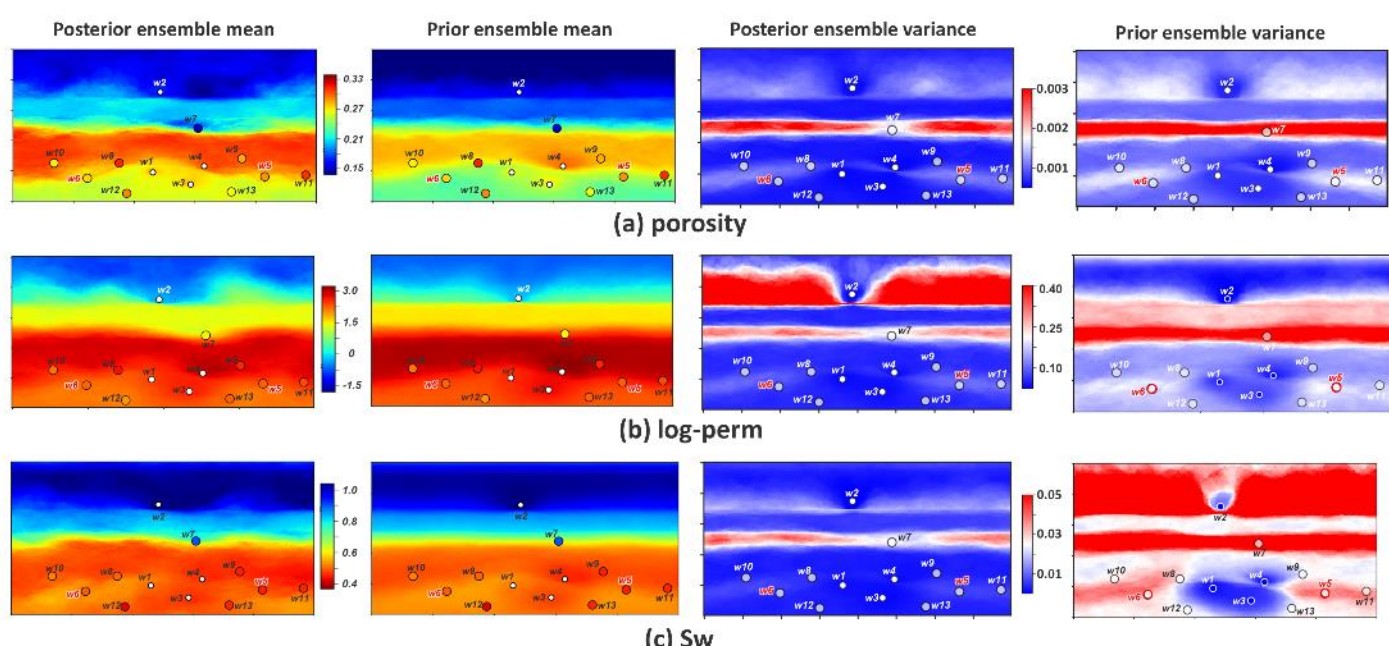

**Figure 18 Ensemble mean and variance of the posterior and prior spatial geological models. The dots represents locations of the boreholes, where color of the dots represents observation values. (a) porosity; (b) log-perm; (c) Water saturation.**



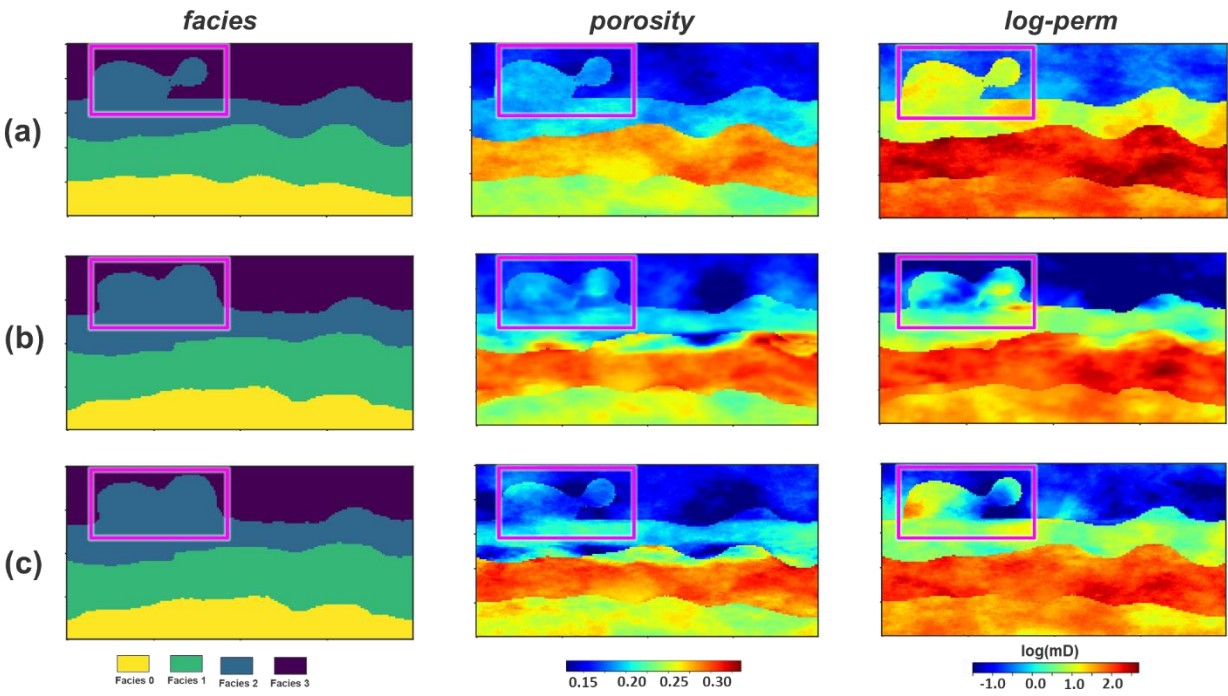

**Figure 19 Prior and posterior facies, porosity and log-perm of realization #1 (a) prior model; (b) posterior model from the sequential decomposition; (c) posterior from joint uncertainty reduction without sequential decomposition.**

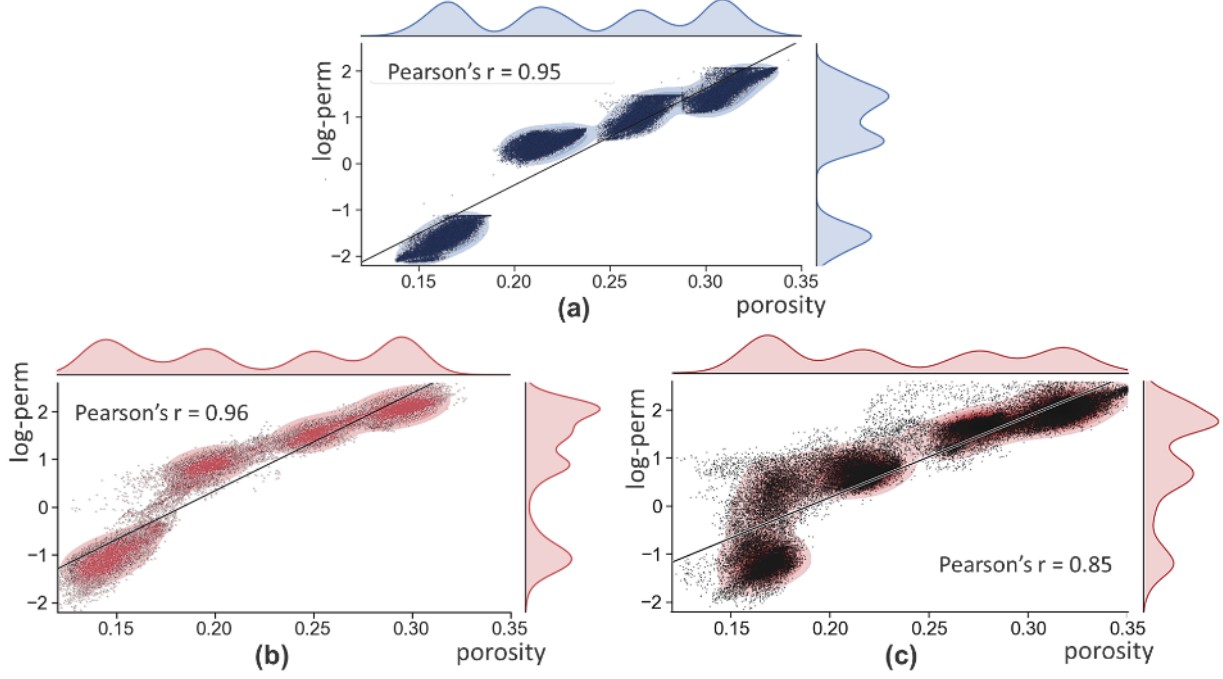

5    **Figure 20.The cross-plots between porosity and log-perm model of realization #1 (a) prior model; (b) posterior model from the sequential decomposition; (c) posterior model without performing the sequential decomposition.**





### 3.4.3 Posterior prediction and falsification

Gas storage volume is calculated using the posterior geological models and plotted in Figure 21. The result highlights a steep uncertainty reduction in comparison to the initial prior prediction. The posterior predicted GIIP leads to a major shift of the expected gas reserve volumes to a more positive direction (higher reserve than initially expected). More importantly, the

5  forecast range is significantly narrowed. This provides critical guidance to the financial decisions on the field development. It also in return confirms the value of the information of the newly drilled wells. In total, the whole application of "Auto-BEL" to this test case took about 45 minutes when running on a laptop with an Intel Core i7 processor and 64 GB of Ram.

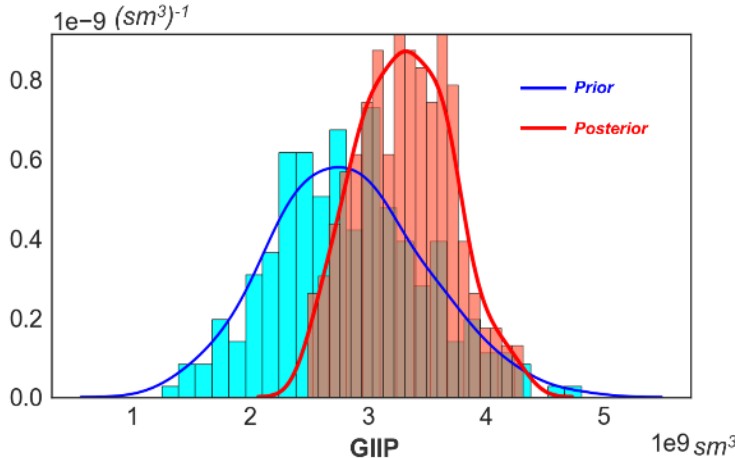

**Figure 21.The prior and posterior prediction of GIIP**

To test the posterior, we perform posterior falsification using data from validating boreholes (w5 and w6). Figure 22 plots the result from applying Robust Mahalanobis Distance outlier detection to the posterior data of the two wells. The statistical test shows that the test borehole observation falls within the main population of data variables, significantly below the 97.5 threshold percentile. We also want to examine if the automated uncertainty reduction can improve posterior model's

15  predictability on the validating boreholes (regarded as future drilling wells). To do so, we compare the prior and posterior predicted thickness at the two borehole locations, together with their actual measurements (Figure 23). For 3D models of facies, porosity, log-perm and Sw, this comparison is performed on vertical average values across the 75 layers. We notice that the posterior model predicts the future borehole observations with significantly reduced uncertainty.





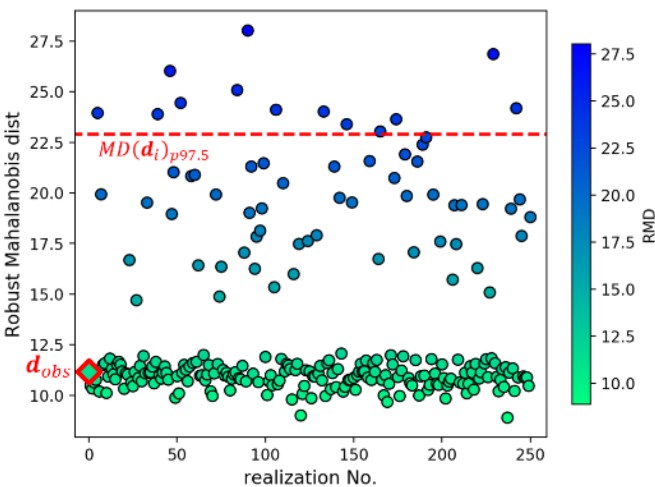

**Figure 22 Posterior falsification using the Robust Mahalanobis Distance outlier detection method using the data from (w5 and w6).**

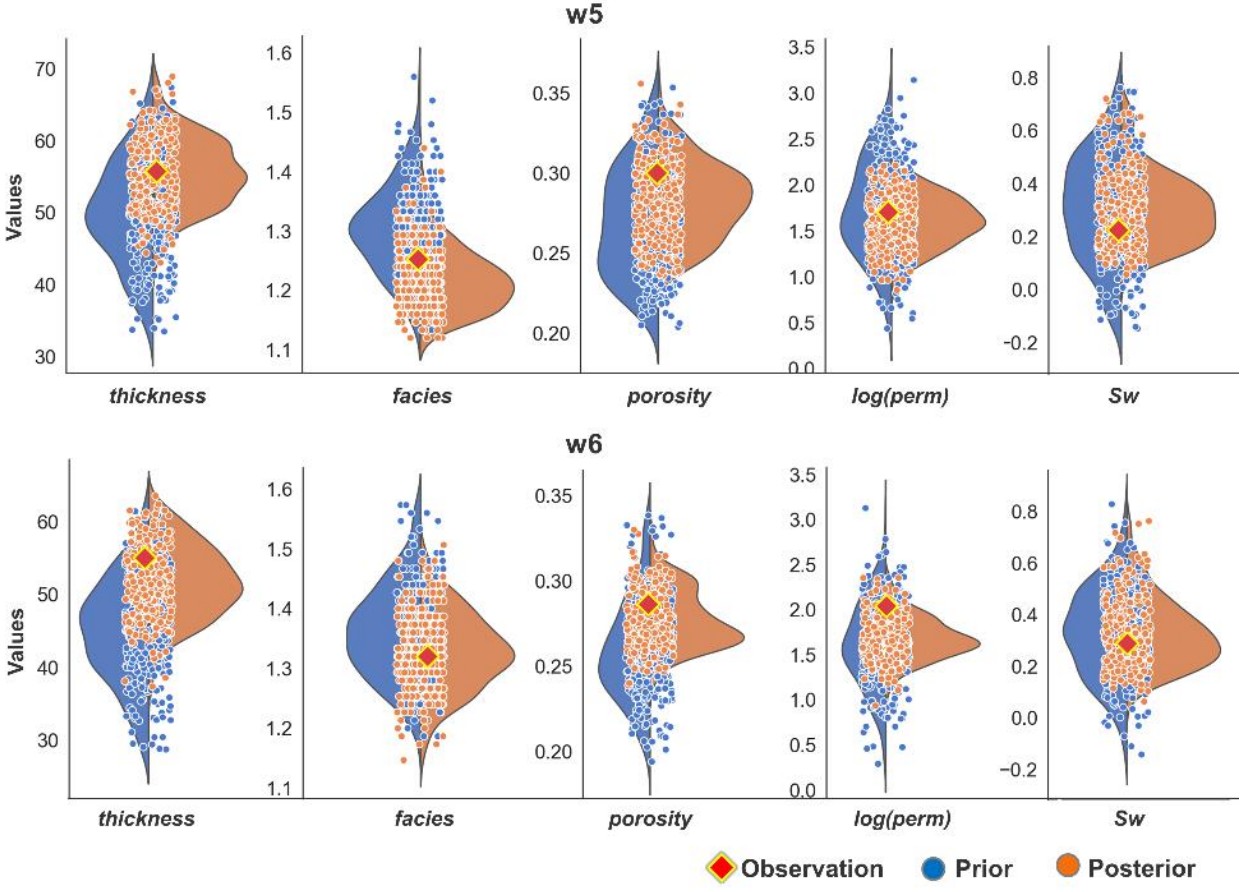

**Figure 23. Prior and posterior predicted thickness, facies, porosity, log-perm and Sw at validating boreholes. The bule and brown**
5  **colored dots respectively represents the prior and posterior prediction, while red squared dots are the actural observations.**





## 4 Discussion and conclusions

In conclusion, we generalized a Monte Carlo-based framework for geological uncertainty quantification and updating. This framework, based on Bayesian Evidential Learning, was demonstrated under the context of geological model updating using borehole data. Within the framework, a sequential model decomposition was proposed, to address the geological rules when

assessing joint uncertainty distribution of multiple model components. For each component, we divided model parameters into global and spatial ones, thus facilitating the uncertainty quantification of complex spatial heterogeneity. When new borehole observations are measured, instead of directly reducing model uncertainty, we first strengthen the model hypothesis by attempting to falsify it via statistical tests. Our second contribution was to show how direct forecasting can jointly reduce model uncertainty under the sequential decomposition. This requires posterior model from previous sequences as additional

inputs to constrain the current prior. Such sequential direct forecasting showed to maintain important geological model features of hierarchy and correlation. In terms of discrete model such as lithology, signed distance function was employed, before applying directly forecasting to reduce uncertainty. As a result, the extension of directly forecasting results an extreme fast computation of posterior geological model, by avoiding conventional model rebuilding. The third contribution, but maybe more important, is that the proposed framework allows automation of geological UQ. We developed an opensource Python

project for this implementation. Its application to a large reservoir model showed that the automated framework ensures the model is objectively informed by data at each step of uncertainty quantitation. It jointly quantified and updated uncertainty of all model components, including structural thickness, facies, porosity, permeability and water saturation. The posterior model showed to be constrained by new borehole observations globally and locally, with dependencies and correlations between the model components preserved from the prior. It predicted validating observations (future drilling boreholes) with reduced

uncertainty. Since posterior cannot be falsified, the uncertainty reduced GIIP prediction can be used for decision makings. The whole process takes less than 1 hour on a laptop workstation for this large field case, thus demonstrating efficiency of the automation

One main purpose of this paper is to introduce automation to geological uncertainty quantification when new borehole data is

acquired. We tackle this challenge by following the protocol of Bayesian Evidential Learning to build an automated UQ framework. BEL formulates a protocol involving falsification, global sensitivity analysis, and data-scientific uncertainty reduction. When establishing such framework for geological UQ, three important questions have to be addressed. The first is on how to preserve the hierarchical relationships and correlations that commonly exist in geological models. We propose a sequential decomposition by following the chain rule under Bayes' theorem. This allows to assess the joint distribution of

multiple model components while honoring the geological rules. The second one is on how to falsify the geological model hypotheses, especially when data becomes highly dimensional. We employ multivariate outlier detection methods. They provide quantitative and robust statistical calculations when attempting to falsify the model using high dimensional data. The last but most practical one, is to deploy data-science driving uncertainty reduction. Uncertainty reduction of geological model is usually time-consuming, because conventional inverse methods require iterative model rebuilding. When it comes to real



cases, the daunting time-consumption and computational efforts of conventional methods can hamper practical implementations of automation. Direct forecasting helps to avoid this, as it mitigates the uncertainty reduction to a linear problem in much lower dimension. However, direct forecasting of geological model is faced with two new challenges. One is to accommodate direct forecasting algorithm to the sequential model decomposition. This is achieved by using the posterior

from the previous sequence as additional constraint. The other challenge is that DF cannot be directly applied to categorical models such as lithological facies. We therefore introduce signed distance function to convert categorical models to continuous properties before performing the DF. Field application has shown benefits of using the proposed framework. Since the posterior in the case study cannot be falsified, its uncertainty can be further reduced by repeating the automated procedures with the validating borehole observations. This suggests that the proposed framework has potentials for life-of-field uncertainty

quantification for applications where new boreholes are regularly drilled.

The main challenge addressed in this paper is to apply such uncertainty quantification within a Bayesian framework. Most method applied in this context simply rebuild the models by repeating the same geostatistical methods that were used to construct the prior model. In such approach, all global variables and their uncertainty need to be re-assessed. The problem with

such approach is twofold. First, it does not address the issue of falsification: the original models may not be able predict the data. Hence, using the same approach to update models with a methodology & prior that may have been falsified may lead again to falsification, thereby leading to invalid and ineffective uncertainty quantification. As a result, the uncertainty quantification of some desirable property, such as volume exhibits a yoyo effect (low variance in each UQ but shifting mean). Second, there is no consistent updating of global model variables. Often such uncertainties are assessed independently of

previous uncertainties. The challenge addressed in this paper is to jointly update global and spatial variables and do this jointly for all properties.

The proposed method offers a Bayesian consistency to uncertainty quantification in the geological modeling setting. However, unlike geostatistical methods, the posterior models do not full match the data at the borehole observations. The current method

is only designed to globally adjust the model, not locally at the borehole observation. One possible path we like to explore in the future is to combine geostatistical conditional simulation as posterior step to the current methodology. A second limitation is that the method does (not yet) treat discrete global variables, such as a geological interpretation. In the cased study, only one interpretation of the lithology was used. The way such variables would be treated is by assigning prior probabilities to each interpretation (e.g. of depositional system), then updating them into posterior probabilities. This has been done by treating the

interpretation independent of other model variables (Aydin and Caers, 2017; Grose et al., 2018; Wellmann et al., 2010). For example, one could first update the probabilities of geological scenarios, then update the other variables. A final, and perhaps more fundamental concern not limited to our approach is on what should be done when the prior model is falsified with new data. According to the Bayesian philosophy this would mean that any of the following could have happened: uncertainty ranges



are too small; model is too simple or some combination of both. The main problem is that it is difficult to assess what the problem is exactly. Our future work will focus on this issue.

## Code availability.

AutoBEL is a free, open-source Python library. It is available at https://doi.org/10.5281/zenodo.3479997, and the source code
is maintained on GitHub https://github.com/sdyinzhen/AutoBEL-v1.0 under MIT license.

## Author contributions

Zhen Yin: contributed the concept and methodology development; wrote and maintained the code; conducted the technical application and drafted this paper.

Sebastien Strebelle: prepared data for the methodology application and provided critical insights during the research
developments.

Jef Caers: provided overall supervision and funding to this project; contributed major and critical ideas to the research development and revised the manuscript.

## Competing interests

The authors declare that they have no conflict of interests.

## Acknowledgements

We thank Chevron for sponsoring this research project. The authors would like to specially thank Maisha Amaru, Jairam Kamath, Lewis Li, Sarah Vitel, Lijing Wang and Celine Scheidt for the technical discussions and supports.

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
