# Peer review of "Automated Monte Carlo-based Quantification and Updating of Geological Uncertainty with Borehole Data (AutoBEL v1.0)"

_Geoscientific Model Development, 2019_

## Referee Comment (RC1) · Anonymous Referee #1 · 19 Nov 2019

Dear authors,

I read with interest your paper entitled "Automated Monte Carlo-based Quantification and Updating of Geological Uncertainty with Borehole Data (AutoBEL v1.0)". This paper presents a new application of the recently developed BEL framework for the updating of reservoir parameters (both local and global parameters) based on borehole data. The main contributions of the paper are (1) the development of a two-step procedure to sequentially predict lithology-related parameters (thickness of the reservoir and facies) and rock properties (permeability, porosity) and (2) a python toolbox with all the necessary code to automatically apply the methodology. The paper is well-structured,

clearly written and deals with important issues related to inverse/prediction problems in Geosciences. I have a series of remarks and suggestions to improve the manuscript. They should be easily handled by slight modifications of the text. Therefore, I suggest publication after minor revision.

General comments

1. The paper claims that the main contribution of the paper is the sequential approach (equation 11). However, how this is actually done is not (sufficiently) explicitly described. In the methodology section, it is written that "we will use the posterior realizations of khi and prior realizations of ksi to determine a conditional distribution f(ksi |khi,posterior), then we evaluate this using borehole observations dobs,ksi of ksi." Later in the manuscript, it is only refer to the use of posterior models as "additional constraints". My understanding is that the posterior distribution of khi represented by 250 realizations gives a new set of thickness and facies. Since the 250 prior realizations of ksi already exist in the input those can be combined to create a new prior, without having to run a new Monte Carlo sampling (what would require to re-run geostatistical realizations). However, the initial realizations of ksi are initially related to other facies distributions. To me, the only way to apply the methodology is then to have a full spatial distribution of porosity and permeability (i.e. covering the entire model) for each facies, so that those can be combined to any facies distribution. I think this point should be clarified and emphasized as it constitutes the core of the methodology.

2. The manuscript lets some ambiguity about the use of synthetic versus field data. I understand that the case is inspired by a field study, but I guess it went through some kind of simplification and that a 251th model of the prior was simulated and considered as the truth ? This has obvious implications on the results, as a synthetic case is always easier to handle (the true model is simulated as part of the prior). I would clarify this point and drop some paragraphs or specific sentences that let think this is actually a field study. For example, section 3.1 is named "the field case", while the first sentence of section indicates that the data set is synthetic. Page 15/Line 12, you

also say that "The actual observations of these data (dobs) are measured from the borehole wireline logs", a clear reference to field data and not simulated data from a synthetic case. However, I assume field wire logs have a higher resolution, so that comparing simulated and field data would require some upscaling ? Similarly, section 3.2.1 contains many reference to data generally collected for oil and gas reservoirs (seismic data) and how the thickness model is deduced from it. This is not necessary as seismic data themselves are not included in the prediction process or the falsification process (see recent paper by Alfonzo and Oliver (2019) for example). Hydrogeological applications would probably use other type of geophysical data for the characterization of those elements (ERT or airborne EM) and removing specific explanations would not reduce the clarity while gaining in generality. If the field data from which the case is inspired were actually used but that the model is considered synthetic because of confidentiality issues, this could simply be stated in the manuscript.

3. Automation is very nice as it probably broadens the potential number of users of the framework, but it comes with potential risks: How do you ensure that intermediate steps remain controlled and within a valid range ? For example, Hermans et al. (2019) show in a case not extremely complex in terms of spatial distribution, that data-prediction (complex) relationships might lead to the impossibility to derive a linear relationship and to apply the linear Gaussian analytical solution (what can be identified in the CCA plots). In such a case, the automated method would give an answer, which would be wrong. Similarly, for the model parameters, the use of the sensitivity analysis provides a way to automatically select the sensitive parameters, but the sensitivity (especially for parameters close to the threshold) might be itself dependent on the number of clusters used for DGSA. For the data, do you use a specific threshold for the automatic selection of the dimension ? How is this threshold related to noise issues ? A short discussion on those issues would help to picture the points where the modeler might need to give some additional input on the inversion process and where further research is needed.

Specific comments

1. Abstract. I recommend to start the abstract with a general sentence on the need for UQ and new methodologies to deal with it, to give some context to the study.

2. Page 6. L8-9. It might be worth mentioning here that the linearization might not be optimal. In such a case alternatives can be to linearize around the observed data, or to use a Kernel density approach (Hermans et al., 2019). Such problem is probably mostly encountered when the link data-prediction is less straightforward than here (borehole data directly measures the model parameters) and include some non-linear forward model.

3. Page 6 L32. Alternatives to PCA to reduce the dimensionality of complex models have been recently proposed such as deep neural network (Laloy et al., 2017, 2018), although they might not be directly applied within the BEL framework.

4. Page 12 L23-29. Maybe I am missing something here. Does the facies also use some secondary variable or trend, such as the thickness or the distance along the Y-axis to be able to represent the belts or is it just the conditioning to borehole data that makes it nicely follow the belt shapes? A simple truncated Gaussian process would rather produce lenses, no?

5. Page 13 L20. Repetition of the section number.

6. Page 16. L11-13. Can you shortly comment in the discussion how AutoBel can be adapted if other type of parameters must be used ?

7. Page 16 L15. I guess that higher order components are somehow sensitive to the initial and the number of realizations and potentially the parameters of the sensitivity analysis. Do you use the 250 components for DGSA ? How much variance is represented by the different components ? Hoffman et al. (2019) in their sensitivity analysis showed that the first 15 PC represented only 23% of the variance, representing the spatial variability with PCA is thus not always straightforward or efficient. Can you comment on that ?

8. Page 17 L2. It is not clear if you apply PCA to the data and what is you threshold on the total variance to select the dimensions ? Is it a user choice or is it fixed in the code ?

9. Figure 12. Not sure to get the color scale for the facies model as the mean is not necessary one of the facies. Maybe show the median, or use a gray scale for one of the facies ?

10. Page 20 L7. I guess the updated uncertainty on porosity, permeability and saturation is performed jointly.

11. Figures 14 and 17. From the sensitivity analysis, it seems that higher order PCs (41, 22, 26, 131) are sensitive for the permeability and they do not correspond to what is shown in Figure 17 (PC1 and PC4).

12. Figure 18. How do you explain that the variance patterns of log-perm and Sw are significantly modified (increase or decrease) after updating in areas where no boreholes are present ? Does not an increase in variance indicate a problem with the prediction, i.e. some predicted parameter values are out of the range of the prior ? The reason could be that the observed data is at the edge of the linear relationship for some component in the CCA.

13. Page 25 L7. I guess the 45 minutes do not include the creation of the 250 MC samples forming the input.

14. Page 28 L30. Lopez-Alvis et al. (2019) recently proposed such an automatic approach for falsification of geological scenarios in a Bayesian hierarchical model based on cross-validation.

References

Alfonzo, M., & Oliver, D.S. 2019. Evaluating prior predictions of production and seismic data. Computer and Geosciences, 23, 1331-1347.

[Figure]

Hoffmann, R., Dassargues, A., Goderniaux, P., Hermans, T., 2019. Heterogeneity and prior uncertainty investigation using a joint heat and solute tracer experiment in alluvial sediments. Frontiers in Earth Sciences - Hydrosphere, Parameter Estimation and Uncertainty Quantification in Water Resources Modeling 7, 108.

Laloy, E., Hérault, R., Jacques, D., Linde, N., 2018. Training-Image Based Geostatistical Inversion Using a Spatial Generative Adversarial Neural Network. Water Resources Research 54, 381–406. https://doi.org/10.1002/2017WR022148

Laloy, E., Hérault, R., Lee, J., Jacques, D., Linde, N., 2017. Inversion using a new low-dimensional representation of complex binary geological media based on a deep neural network. Advances in Water Resources 110, 387–405. https://doi.org/10.1016/j.advwatres.2017.09.029

Lopez-Alvis, J., Hermans, T., Nguyen, F., 2019. A cross-validation framework to extract data features for reducing structural uncertainty in subsurface heterogeneity. Advances in Water Resources 133, 103427.

---

## Short Comment (SC1) · 6 Dec 2019

Dear authors,

This paper presents an automated workflow to build geo-model using some hard data. The work is very interesting and the paper is well written. I strongly suggest publication after some revision. I have a few questions and comments. I hope authors can clarify it.

1. In BEL, you mentioned prediction. There is nothing related prediction from the field case, am I right? It seems to me this work is mainly related model building using data.

[Figure]

The prediction variable h is the model parameters, right?

2. Using observation d_obs, you can detect some outlier realizations. Do you just remove these realizations from your prior in practice?

3. h*, d* are some subspace of h and d, right? If we talking permeability field with millions of cells, could you give me roughly number of h* compared to h? Is Formula (9) standard way to formula linear-Gauss problem?

4. Your Python tool can be used to build geo-mode l(grid, etc..)?

5. After CCA, elements in h* is more independent (less correlated), right?

6. I am very interested in "sequential update model". It will be nice if the authors can describe this in more details.

7. Every time you update a parameter, do you use the posterior as a prior for next parameter update?

---

## Short Comment (SC2) · 16 Dec 2019

Geological models can be more accurate and actual with coupling more borehole data in models. Meanwhile, the data sizes of geological models increase with the developments of field projects and participation of new borehole data. During dynamic process of subsurface applications such as groundwater, geothermal, oil, gas, and CO2 geo-storage, uncertainty quantification is the key for decision making. As the authors mentioned, uncertainty reduction is a time-consuming work which requires iterative model rebuilding using conventional inverse methods. In order to make the model adhere to geological rules, geological modeling often requires significant individual/group expertise and manual intervention which will need often months of work after new data is achieved. In this paper, the authors generalized a Monte Carlo-based framework for geological uncertainty quantification and updating. Their methodologies were developed with the BEL protocol for uncertainty quantification. The extension of directly forecasting results an extreme fast computation of posterior geological model, by avoiding conventional model rebuilding. The proposed framework also allows automation of geological UQ. This paper is interesting and in an area worthy of investigation. Overall, this paper is well-organized and well-written. This paper can be accepted by addressing the following minor comments.

1. The advantages and disadvantages using your method for UQ and updating should be further illustrated by comparing with typical conventional method. At the same time, its applicable scenarios are suggested to be provided which can give guidelines for field application.

2. As you mentioned, current method is only designed to globally adjust the model, not locally at the borehole observation. Could you provide your idea on further solution in more details?

3. Could provide the specific performance parameters of CPU which can show the improvement on calculation efficiency more accurately?

4. The authors are suggested to unify the multiplication sign through the whole manuscript?

5. Please add a "." between "Figure 19" and the "Prior and posterior..." to keep in accordance with other figures. Please check similar problems accordingly.

6. The usage of abbreviation such as DF should be noticed. Abbreviations should be defined when they are first mentioned in the text and should always be used afterwards.

7. Discussion and conclusions are suggested to be separated into two parts. Please provide conclusions point by point which can help reader to understand the main contributions of the paper. Meanwhile, future researches should be clarified according to the limitations of proposed method.

---

## Referee Comment (RC2) · Guillaume Caumon (Referee) · 27 Dec 2019

**General comments**

This paper proposes an application of the "Bayesian Evidential Learning" approach to the problem of uncertainty assessment in integrated reservoir modeling. Although the general approach was previously described in Scheidt et al (2018), this paper contains significant new elements, applications and discussions, which are very interesting for the community.

In terms of form, the paper is very well written and clearly presented, apart from minor

issues. It includes a link to a Jupyter notebook implementing the methodology and applying it to the reservoir thickness. The implementation works fine, after some twidling to install scikit-fmm. The overall structure of the code seems to allow for managing facies (some unused functions for facies modeling are present in the code), but the demo notebook assumes porosity to be 1 and water saturation to be zero. Even so, the reproducibility is much better than in most similar papers on this topic.

Overall, I congratulate the authors for the very interesting approach which represents a paradigm shift as compared to the current practice. I have, nonetheless, several comments, questions, and suggestions, which I hope will help to improve the paper. My recommendation is to proceed with minor/moderate revision.

**Specific comments**

- Overall, the introduction makes a good job introducing the general problem, but more precise explanations about the exact contributions of the paper would be welcome at this stage (in particular with regard to the other recent contributions of SCREF).
- The borehole data are generally at much higher resolution than the reservoir grid data. However, as in most reservoir modeling approaches, this paper assumes that the borehole data has been upscaled to grid resolution, a process which is source of inaccuracies in reservoir models. One of the points of the proposed approach is that the falsification step (Section 2.1.3) could in principle integrate the scale change. Comments on this would be welcome. Also, some additional precisions about the management of categorical variables during falsification would be welcome (in addition to the last sentence on page 5). From Eq. (8), it seems that the robust Mahalanobis distance accounts for spatial redundancy; please confirm.
- Overall, I am not fully clear about the falsification step. As this is a key aspect

of the proposed methodology, it would be good if the authors could insist on the consequences of this approach as compared to the conventional method which creates models sampling exactly the borehole data. Errors between model and data may be acceptable for some applications such as hydrocarbon in place, as they will average out, but would they yield reliable forecasts of porous flow and transport if borehole data are not exactly honored by some model realizations?

- Although this is not the main point of the paper, some parameters for generating the prior models could be described more precisely and discussed. For example, I am a bit puzzled by uniform distributions taken for the facies 1,2,3, which suggests that facies 0 will adjust so that the total is equal to 1, which may be a source of bias (see Haas Formery, Math. Geol. 2002, or compositional data analysis literature). What are the variogram ranges for facies modeling? In the figure, there seems to be a facies trend, but what are the parameters of this trend? Are the variogram models isotropic? Is the variogram of porosity the same for all facies?

- Overall, I get the overall idea for facies, but I am not fully clear about the consequence of the facies processing. The signed distance is mentioned, and then the Truncated Gaussian. I first understood that the TG was used in the generation of prior models, and the signed distance for the workflow steps (which would mean in general 3 distance fields for 4 facies). But I was then puzzled by Fig 12 which suggests that maybe one single scalar field is used. So at the end, I am not sure about what was done exactly. Clarification of this would be needed in the paper.

- I looked at the code to try to understand the facies management, but it is not fully integrated in the high-level functions. Adding facies management in the code would improve reproducibility. If not possible, then please explicitly mention in Section 2.3 that the provided python code illustrates the workflow for thickness only.

- I have some doubts about the back transformation process when not enough

PCs are retained. Some artifacts are visible in the realizations on Fig. 19 and on the poro/perm plots of Fig. 20 (breaking the consistency between petrophysical features within each facies). Could this also break the match to borehole data if not enough PCs were selected?

- As the aim of the paper is to “minimize the need for tuning parameters” it would be good to summarize the updated parameters as in Table 1 to help the reader assess to what extent the global model parameters have been updated by the process. I cannot help but wondering about how the updating of global parameters such as facies proportions or variogram range would compare to a classical process where statistical inference would be repeated by domain experts as new data become available.
- Performance: I assume the 45 minutes do not include the prior model generation? Please clarify.
- Discussion: the discussion in its current form highlights the main points of the method and some challenges, but does not really discuss some aspects which are often considered critical in subsurface models, such as the match of individual realizations to borehole data, or the preservation “geological consistency” such as the petrophysical distributions for various rock types. I suspect some moderate violations do not really matter for the accumulation problem considered in this paper, but I have more doubts about what would happen for modeling objectives involving highly non-linear physics, such as flow simulation. SOme balanced discussion on these aspects would probably be useful. Another question is whether there are any guidelines about the various sensitivity / confidence levels involved in the method, as these parameters likely impact the results.

**Technical corrections**

In several places: the term "data-scientific" looks hype but I don't get the exact meaning. Please define or use another term.

Also, the term “constraint” used in several places could probably be replaced with more accurate terminology.

p.1, l.19: “A generalized synthetic data set motivated by a gas reservoir”: please rephrase. Seems to me this is a gas reservoir study which has been simplified.

p.4, Eq. (3): Unless I am missing something, the notation could be simpler using $\chi_{sp}, \chi_{gl}$, etc.

p.6, l.10: Please define $G$ (and make it bold?). Fix typos in Eq. (10): “proir” should read prior.

p.6, l.24-25: Please rephrase the sentence for more clarity. This is more an explanation about why it works in practice than an actual “truth”.

p.7, l.1: Instead of “model grid cells”, I'd suggest for generality: model parameters

p.7, l.3-6: I'd recommend to factually summarize the DGSA approach rather than summarizing its advantages (as compared to what?).

p.7, l.11-20: The notations of Eq 11 are unclear to me. I get the point of the sequential updating, but I am not sure it is correct to write “the posterior model of $\chi$ becomes prior model for ”, as both variables are different. Maybe using the subscript such as $\chi_{posterior}$ in Eq (11) would help make the point clearer.

P.7, l. 21-25: PCA on an image can be achieved in a variety of manners. I suspect that in Figs. 1 and 2, the PCA factors are linear combinations of image columns, but please explain so that the reader does not have to guess.

P.7, l. 27: I'd suggest to use $\psi_s$, as it relates to the distance to facies $s$. I also think that $\mathbf{x}_\beta$ should be the closest location equal to (and not different from)$s$ in the second term (otherwise, the definition enlarges the facies by 1 voxel). Or maybe just the distance to

the boundary of facies $s$?

p.8, l. 7: Typo: the prior uncertainty models *have*

p.9, l.1-3: Please check convoluted sentence.

p.9, l.13: Did you mean “the reservoir rocks deposited at shallow marine environments”?

p.10, l.2: Confusion between in-place resources (GIIP) and recoverable reserves, which also depends on flow behavior.

p.10, l.12-13: Please check syntax.

P.10, Fig.5: a scale would be needed so that variogram ranges provided later in the paper can be related to model size and well spacing.

p.11, table 1: Typo in “gammy ray”

p.12, l.2: In addition to resolution, velocity is a significant source of thickness uncertainty.

p.13, l.10-15: $h$ is the height above the free water level, not the reservoir depth.

p.13, l.19-20: Please fix sentence.

p.16, l.11: The independence between thickness and facies is stated as a fact. In my view, it is a working hypothesis (probably a reasonable one), but not a general truth.

p. 18, Fig.10: Would $d_{obs}$ correspond to a line? I guess it should have some thickness due to data noise and to PCa projection.

p.19, l.1-2: “the uncertainty... their prior”: Please chech grammar.

p.19, Fig.11: Please tell what the dash lines correspond to (kernel density?)

p.20, Fig. 12: The visualization for posterior facies distributions gives a qualitative hint about what happens, but I am not sure about what we are exactly looking at. It is the

blending of discrete color maps or the average of the underlying Gaussian field (but then, facies threshold change depending on facies proportions)?

Fig. 15: Typo in legend

p.25: Please remove mention to reserves, as no recovery factor is involved.

p.25, l.17: The cross-validation tests on wells 5 and 6 seems a bit optimistic, as vertical averaging on the 75 layers essentially amounts to making a two-dimensional model. Again, erors average out, so the reduction of uncertainty in such a case is no proof of the actual forecasting ability of the model in three dimensions.

p. 24, Fig. 20: The correlation coefficient is not really meaningful on such multi-modal distributions, even more so as the facies proportions likely change from one distribution to another. It would make more sense to examine the bivariate statistics per facies.

P.26, Fig. 23: please explain what we see: the curves are densities, but what does the point horizontal spread mean? And again, what is the facies value?

p.27 l.12, l.15, l.24 : “results an extreme fast computation”; “be able predict”, “do not full match”: Please fix English

p.28, L.11-21: This paragraph nicely explains the problem in simple terms, so I think it would be better integrated in the introduction than in the discussion.

p.28, L.16: this sentence mixes the falsification of parameters and falsification of a methodology, which I think are very different. I suggest rephrasing.

p.28, l.30: Not sure I understand the references in this context.

---

## Author Comment (AC1) · 10 Jan 2020

Dear Referee,

Thanks for taking time to review our paper "Automated Monte Carlo-based Quantification and Updating of Geological Uncertainty with Borehole Data (AutoBEL v1.0)". We highly appreciate your comments and suggestions. Please find blew our responses and manuscript revisions. The provided responses below include 3 part:

- Part 1. Response to your comments (RC1).
- Part 2. Revised manuscript based on your and other referee's comments.
- Part 3. Responses to comments from other referees (RC2, SC1, SC2).

The responses and manuscript revisions are highlighted in red.

We are looking forward to a positive evaluation.

On behalf of the Authors,
Zhen Yin (David)

**Responses to Anonymous Referee #1 on interactive comments (RC1)**

Dear authors,

I read with interest your paper entitled "Automated Monte Carlo-based Quantification and Updating of Geological
Uncertainty with Borehole Data (AutoBEL v1.0)". This paper presents a new application of the recently developed
BEL framework for the updating of reservoir parameters (both local and global parameters) based on borehole
data. The main contributions of the paper are (1) the development of a two-step procedure to sequentially predict
lithology-related parameters (thickness of the reservoir and facies) and rock properties (permeability, porosity)
and (2) a python toolbox with all the necessary code to automatically apply the methodology. The paper is well-
structured, clearly written and deals with important issues related to inverse/prediction problems in Geosciences.
I have a series of remarks and suggestions to improve the manuscript. They should be easily handled by slight
modifications of the text. Therefore, I suggest publication after minor revision.

We highly appreciate the in-depth review and constructive suggestions from the reviewer. Overall, we agree with
the reviewer comments and these have improved the clarity of our paper. Provided below are our point-to-point
responses and explains how they are addressed in the revised manuscript.

**General comments**

1. The paper claims that the main contribution of the paper is the sequential approach (equation 11). However,
how this is actually done is not (sufficiently) explicitly described. In the methodology section, it is written that "we
will use the posterior realizations of khi and prior realizations of ksi to determine a conditional distribution
f(ksi|khi,posterior), then we evaluate this using borehole observations dobs,ksi of ksi." Later in the manuscript, it
is only refer to the use of posterior models as "additional constraints". My understanding is that the posterior
distribution of khi represented by 250 realizations gives a new set of thickness and facies. Since the 250 prior
realizations of ksi already exist in the input those can be combined to create a new prior, without having to run a
new Monte Carlo sampling (what would require to re-run geostatistical realizations). However, the initial
realizations of ksi are initially related to other facies distributions. To me, the only way to apply the methodology
is then to have a full spatial distribution of porosity and permeability (i.e. covering the entire model) for each
facies, so that those can be combined to any facies distribution. I think this point should be clarified and
emphasized as it constitutes the core of the methodology.

Yes, the reviewer's understanding of the sequential approach is correct and exactly what we proposed in the
paper. The prior Monte Carlo model samples provide a full spatial distribution of all the model components,
including porosity and permeability for each facies. This allows the calculation of posterior porosity and
permeability models to fit any (posterior) facies distributions using direct updating. Therefore, we use the
posterior facies model and borehole porosity observations as constraint in AutoBEL to calculate the posterior
porosity.

We have added a paragraph to the application section (section 3.4.2) to explicitly explain how the sequential approach is performed.

2. The manuscript lets some ambiguity about the use of synthetic versus field data. I understand that the case is inspired by a field study, but I guess it went through some kind of simplification and that a 251th model of the prior was simulated and considered as the truth? This has obvious implications on the results, as a synthetic case is always easier to handle (the true model is simulated as part of the prior). I would clarify this point and drop some paragraphs or specific sentences that let think this is actually a field study. For example, section 3.1 is named

"the field case", while the first sentence of section indicates that the data set is synthetic. Page 15/Line 12, you also say that "The actual observations of these data (dobs) are measured from the borehole wireline logs", a clear reference to field data and not simulated data from a synthetic case. However, I assume field wire logs have a higher resolution, so that comparing simulated and field data would require some upscaling? Similarly, section 3.2.1 contains many reference to data generally collected for oil and gas reservoirs (seismic data) and how the thickness model is deduced from it. This is not necessary
as seismic data themselves are not included in the prediction process or the falsification process (see recent paper by Alfonzo and Oliver (2019) for example). Hydrogeological applications would probably use other type of geophysical data for the characterization of those elements (ERT or airborne EM) and removing specific explanations would not reduce the clarity while gaining in generality. If the field data from which the case is inspired were actually used but that the model is considered synthetic because of confidentiality issues, this could simply be stated in the manuscript.
Yes, the field data for prior modeling and posterior calculation are from a real case. But the model is considered as synthetic because of simplification for generic application, and because of confidential issues. There is no 251st model of the prior to be considered as the truth. We add a statement for this in the revised manuscript.

The actual well observations are in grid model resolution already. They are upscaled to from higher resolution wireline logs. Unfortunately, we don't have the original high resolution well logs due to confidential concerns. We have rephrased Page15/Line12 in the revision to avoid the confusions.

In section 3.2.1, we have rephrased and shortened the detailed seismic explanations to gain more generality while keeping the clarity.

3. Automation is very nice as it probably broadens the potential number of users of the framework, but it comes with potential risks: How do you ensure that intermediate steps remain controlled and within a valid range? For example, Hermans et al. (2019) show in a case not extremely complex in terms of spatial distribution, that data-prediction (complex) relationships might lead to the impossibility to derive a linear relationship and to apply the linear Gaussian analytical solution (what can be identified in the CCA plots). In such a case, the automated method would give an answer, which would be wrong. Similarly, for the model parameters, the use of the sensitivity analysis provides a way to automatically select the sensitive parameters, but the sensitivity (especially for parameters close to the threshold) might be itself dependent on the number of clusters used for DGSA. For the
data, do you use a specific threshold for the automatic selection of the dimension? How is this threshold related to noise issues? A short discussion on those issues would help to picture the points where the modeler might need to give some additional input on the inversion process and where further research is needed.

The intermediate steps of automation are ensured by prior/posterior falsifications and sensitivity analysis. Prior falsification guarantees the uncertainty distribution range of prior model is not wrong. Posterior falsification
validates the calculated posterior model. The use of sensitivity analysis ensures sensitive model parameters are informed by the data during uncertainty reduction. Besides, the intermediate steps can also be adjusted for users' specific application problems to ensure that the results are meaningful.

We know here for a fact that the forward problem is linear. The borehole data variable (simulated borehole data)
are extracted from the model, which is simply a matrix operation. This is shown in Eq.4. Hence the CCA approach will always work here. For more complex non-linear inverse problems as mentioned in Hermans et al. (2019), statistical estimation approaches such as kernel density estimation can be used to replace CCA and Gaussian regression steps, and there are also extensions of CCA to tackle non-linear problems (e.g., Lai and Fyfe, 1999). We have extended the discussion section to address this problem. However, the paper only deals with borehole data

We agree that the cluster number used in DGSA can influence the calculation of sensitive model parameters. The effects of cluster number has been thoroughly studied by Park et al. (2016). We won't repeat this in the manuscript, but we have cited this paper in section 2.2.1. Here we use 3 clusters in DGSA as we focus on the main sensitivity effects.

Regarding the data variable, we select PCs number by preserving 90% variance. This is first because borehole data are in much lower dimension than the spatial models. It does not require much further dimension reduction. Besides, preserving 90% of the variance can be useful to filter out higher order PCs which may relate to local noises. We didn't consider the influence of noise on the threshold. We added more explanations to section 2.2.1
further explain the dimension reduction of model and data variables.

References:

Lai, P., and Fyfe, C.: A neural implementation of canonical correlation analysis, Neural Networks, 12(10), 1391–1397, doi:10.1016/S0893-6080(99)00075-1, 1999.

Lopez-Alvis, J., Hermans, T. and Nguyen, F.: A cross-validation framework to extract data features for reducing structural uncertainty in subsurface heterogeneity, Adv. Water Resour., 133, 103427, doi:10.1016/J.ADVWATRES.2019.103427, 2019.

Park, J., Yang, G., Satija, A., Scheidt, C. and Caers, J.: DGSA: A Matlab toolbox for distance-based generalized sensitivity analysis of geoscientific computer experiments, Comput. Geosci., 97, 15–29, doi:10.1016/J.CAGEO.2016.08.021, 2016.

**Specific comments**

1. Abstract. I recommend to start the abstract with a general sentence on the need for UQ and new methodologies
to deal with it, to give some context to the study.

We have added a general sentence to start the abstract.

2. Page 6. L8-9. It might be worth mentioning here that the linearization might not be optimal. In such a case alternatives can be to linearize around the observed data, or to use a Kernel density approach (Hermans et al., 2019). Such problem is probably mostly encountered when the link data-prediction is less straightforward than
here (borehole data directly measures the model parameters) and include some non-linear forward model.

We added a discussion on the limitations of linearization in the discussion section, citing the reference of Hermans et al., 2019.

3. Page 6 L32. Alternatives to PCA to reduce the dimensionality of complex models have been recently proposed
such as deep neural network (Laloy et al., 2017, 2018), although they might not be directly applied within the BEL framework.

Thanks for providing the new approaches. The two references are added to the revision to suggest the alternative dimension reduction methods. But here we recommend using PCA for its bijectivity, and simplicity. For more complex geological settings one may indeed need to rely on non-linear techniques.

4. Page 12 L23-29. Maybe I am missing something here. Does the facies also use some secondary variable or trend, such as the thickness or the distance along the Yaxis to be able to represent the belts or is it just the conditioning to borehole data that makes it nicely follow the belt shapes? A simple truncated Gaussian process would rather produce lenses, no?

Yes, a geometrical trend was applied to prior facies modelling using truncated Gaussian. This is to make sure that the orientation and position of modeled facies belts are consistent with geological interpretations in Figure 5(a). The trend in the case study was provided by the field geologists with comprehensive field geology studies, and officially used by the field operator. We therefore quantify the facies model uncertainty under this trend scenario.

5. Page 13 L20. Repetition of the section number.

Section number has been corrected.

6. Page 16. L11-13. Can you shortly comment in the discussion how AutoBel can be adapted if other type of parameters must be used?

Generally, AutoBEL is easy to adapt if the other types of parameters are used for uncertainty quantification. This can be done by simply adding the additional parameters to the model variable **m**.

We added a short comment on this in the discussion section.

7. Page 16 L15. I guess that higher order components are somehow sensitive to the initial and the number of realizations and potentially the parameters of the sensitivity analysis. Do you use the 250 components for DGSA? How much variance is represented by the different components? Hoffman et al. (2019) in their sensitivity analysis showed that the first 15 PC represented only 23% of the variance, representing the spatial variability with PCA is thus not always straightforward or efficient. Can you comment on that?

Yes, we used all the 250 model components for DGSA. The sensitive PCs of thickness and facies model represents about 19% of total variance. We don't think this is too small or not efficient, it simply a direct result from the fact that for our case, borehole data are in very low dimension with only 7 locations for such a large field. They cannot inform many components of such large dimensional models.

We agree that it is important to know the variance represented by sensitive PCs. This is helpful to understand how informative the boreholes are (depending on their locations). We added Hoffman et al. (2019) to our references.

8. Page 17 L2. It is not clear if you apply PCA to the data and what is you threshold on the total variance to select the dimensions? Is it a user choice or is it fixed in the code?

We select PCs of data by preserving 90% variance (fixed in this code). This is first because borehole data are in much lower dimension than the spatial models. It does not require much further dimension reduction. Besides, preserving 90% of the variance can be useful to filter out higher order PCs which may relate to local noises. We clarified this in the new revision.

9. Figure 12. Not sure to get the color scale for the facies model as the mean is not necessary one of the facies. Maybe show the median, or use a gray scale for one of the facies?

We have replaced the mean by median model

10. Page 20 L7. I guess the updated uncertainty on porosity, permeability and saturation is performed jointly.

Yes, they are updated jointly under the sequential decomposition. We added a paragraph to explain how the joint update is performed.

11. Figures 14 and 17. From the sensitivity analysis, it seems that higher order PCs(41, 22, 26, 131) are sensitive for the permeability and they do not correspond to what is shown in Figure 17 (PC1 and PC4).

In Figure17, we plot two sensitive PCs with highest variances. For permeability, higher order PCs (41, 22, 26, 131) are more sensitive than PC1 and PC4, but PC1 and PC4 contain more information than those higher PCs. Therefore, we choose to plot PC4 and PC1.

We rephrased the figure caption for more clarity.

12. Figure 18. How do you explain that the variance patterns of log-perm and Sw are significantly modified (increase or decrease) after updating in areas where no boreholes are present? Does not an increase in variance indicate a problem with the prediction, i.e. some predicted parameter values are out of the range of the prior? The reason could be that the observed data is at the edge of the linear relationship for some component in the CCA.

Well spotted! Sorry, the prior variance figures of log-perm and Sw are misplaced. In fact, the prior variance figure of log-perm in Figure 18(b) is for Sw, and vice versa. We corrected this mistake in Figure 18. The new comparison between the prior and posterior does not show the significantly increase/decrease of variances. They now are the correct results of uncertainty reduction in log-perm and Sw.

13. Page 25 L7. I guess the 45 minutes do not include the creation of the 250 MC samples forming the input.
Correct. The time used for creating the 250 prior samples are not counted. The 45 minutes are for prior falsification, sensitivity analysis, geological uncertainty reduction with new borehole observations, and posterior falsification. We clarified this in the revision.

14. Page 28 L30. Lopez-Alvis et al. (2019) recently proposed such an automatic approach for falsification of geological scenarios in a Bayesian hierarchical model based on cross-validation.
We added this reference to the revision.

References

Alfonzo, M., & Oliver, D.S. 2019. Evaluating prior predictions of production and seismic data. Computer and Geosciences, 23, 1331-1347.

Hoffmann, R., Dassargues, A., Goderniaux, P., Hermans, T., 2019. Heterogeneity and prior uncertainty investigation using a joint heat and solute tracer experiment in alluvial sediments. Frontiers in Earth Sciences - Hydrosphere, Parameter Estimation and Uncertainty Quantification in Water Resources Modeling 7, 108.

Laloy, E., Hérault, R., Jacques, D., Linde, N., 2018. Training-Image Based Geostatistical Inversion Using a Spatial Generative Adversarial Neural Network. Water Resources Research 54, 381–406. https://doi.org/10.1002/2017WR022148

Laloy, E., Hérault, R., Lee, J., Jacques, D., Linde, N., 2017. Inversion using a new low dimensional representation of complex binary geological media based on a deep neural network. Advances in Water Resources 110, 387–405. https://doi.org/10.1016/j.advwatres.2017.09.029

Lopez-Alvis, J., Hermans, T., Nguyen, F., 2019. A cross-validation framework to extract data features for reducing structural uncertainty in subsurface heterogeneity. Advances in Water Resources 133, 103427.

Thanks for providing the references. They have been added to improve the paper.

**Part 2. Revised manuscript:**

[revised manuscript text omitted]

**Part 3.**

**Responses to Guillaume Caumon (Referee) on interactive comments (RC2)**

**Guillaume Caumon (Referee)**

guillaume.caumon@ensg.univ-lorraine.fr

**General comments**

This paper proposes an application of the "Bayesian Evidential Learning" approach to the problem of uncertainty assessment in integrated reservoir modeling. Although the general approach was previously described in Scheidt et al (2018), this paper contains significant new elements, applications and discussions, which are very interesting for the community.

In terms of form, the paper is very well written and clearly presented, apart from minor issues. It includes a link to a Jupyter notebook implementing the methodology and applying it to the reservoir thickness. The implementation works fine, after some twiddling to install scikit-fmm. The overall structure of the code seems to allow for managing facies (some unused functions for facies modeling are present in the code), but the demo notebook assumes porosity to be 1 and water saturation to be zero. Even so, the reproducibility is much better than in most similar papers on this topic.

Overall, I congratulate the authors for the very interesting approach which represents a paradigm shift as compared to the current practice. I have, nonetheless, several comments, questions, and suggestions, which I hope will help to improve the paper. My recommendation is to proceed with minor/moderate revision.

We are very thankful to the referee's thorough and in-depth review. The comments almost cover all the aspects of our paper. They are insightful and very helpful to improve our work. We overall agree with them. Provided below shows how the referee's comments are addressed point-to-point and incorporated to the revised manuscript

**Specific comments**

• Overall, the introduction makes a good job introducing the general problem, but more precise explanations about the exact contributions of the paper would be welcome at this stage (in particular with regard to the other recent contributions of SCREF).

We rephrased the last paragraph of "Introduction" to clarify the exact contributions of our paper with regard to the previous papers on this topic (of BEL).

The main contributions are

- To propose a model falsification scheme using robust Mahalanobis distance.

- Extension direct forecasting based on sequential model decomposition to honor the hierarchical rules in geological modeling.
- A complete automation of geological uncertainty quantification using borehole data.

• The borehole data are generally at much higher resolution than the reservoir grid data. However, as in most reservoir modeling approaches, this paper assumes that the borehole data has been upscaled to grid resolution, a process which is source of inaccuracies in reservoir models. One of the points of the proposed approach is that the falsification step (Section 2.1.3) could in principle integrate the scale change. Comments on this would be welcome. Also, some additional precisions about the management of categorical variables during falsification would be welcome (in addition to the last sentence on page 5). From Eq. (8), it seems that the robust Mahalanobis distance accounts for spatial redundancy; please confirm.

In this paper we deal with borehole observations that are already upscaled to model grid resolution. Upscaling errors is not within the scope of this paper's research, but is for us a very active area of our current research. The way we approaching this problem is to add such error to the data covariance matrix in Eq10, using some Monte

Carlo analysis, but as said, see our next paper. As a result the falsification then indeed tests only a necessary condition (upscaled data) but not a sufficient test (actual data)

One rather ad-hoc trick around that is to choose a lower threshold value as the tolerance of Chi-squared distribution of $\mathrm{RMD}(\mathbf{d})$ (making the test more powerful) We have added a few sentences in Section 2.1.3 to further clarify this problem.

Yes, spatial redundancy is accounted by performing dimension reduction (PCA) on the well data. Eq (8) uses the PC scores of $\mathbf{d}$.

• Overall, I am not fully clear about the falsification step. As this is a key aspect of the proposed methodology, it would be good if the authors could insist on the consequences of this approach as compared to the conventional method which creates models sampling exactly the borehole data. Errors between model and data may be acceptable for some applications such as hydrocarbon in place, as they will average out, but would they yield reliable forecasts of porous flow and transport if borehole data are not exactly honored by some model realizations?

The goal of falsification is not to check errors between model and data, but to check whether the model can predict the data, even if we would account for some tolerance (or error). If that is not the case, then one would be averaging out over a wrong model. Therefore, we are not looking here at variance of error, but bias in the prior

• Although this is not the main point of the paper, some parameters for generating the prior models could be described more precisely and discussed. For example, I am a bit puzzled by uniform distributions taken for the facies 1,2,3, which suggests that facies 0 will adjust so that the total is equal to 1, which may be a source of bias (see Haas Formery, Math. Geol. 2002, or compositional data analysis literature). What are the variogram ranges
for facies modeling? In the figure, there seems to be a facies trend, but what are the parameters of this trend? Are the variogram models isotropic? Is the variogram of porosity the same for all facies?

Yes, the prior modeling is not the focus of our paper. We therefore prefer not to stress much on the prior modeling, but to focus more on our main contributions on falsification, direct forecasting on sequential decomposition and automation. Additionally, Referee #1 suggests to remove specific explanations on the prior
modeling to gain more generality (see General comment #2 of Referee #1). We agree with this comment.

Regarding the prior facies modeling, a deterministic trend was applied. This is to make sure that the orientation and position of modeled facies belts are consistent with geological interpretations in Figure 5(a). The trend in the case study was provided by the field geologists, and officially used by the field operator. The variogram models
are isotropic for facies, anisotropic for porosity. The variogram of porosity is the same in all facies. We quantify the prior model uncertainty under this scenario.

• Overall, I get the overall idea for facies, but I am not fully clear about the consequence of the facies processing. The signed distance is mentioned, and then the Truncated Gaussian. I first understood that the TG was used in the generation of prior models, and the signed distance for the workflow steps (which would mean in general 3
distance fields for 4 facies). But I was then puzzled by Fig 12 which suggests that maybe one single scalar field is used. So at the end, I am not sure about what was done exactly. Clarification of this would be needed in the paper.

The referee is correct on understanding the overall idea for facies uncertainty quantification. We agree that the main problem is from Fig 12 which is confusing. We replotted Fig 12 to avoid such confusions. The figure is replaced by the median posterior facies model to show the final results from Auto-BEL, which is also suggested
by Referee #1.

• I looked at the code to try to understand the facies management, but it is not fully integrated in the high-level functions. Adding facies management in the code would improve reproducibility. If not possible, then please explicitly mention in Section 2.3 that the provided python code illustrates the workflow for thickness only.

We appreciate the referee's test of the Auto-BEL code. Regarding the problem of facies management, we couldn't
provide the facies model data because it is classified as confidential by the company. However, all the code functions for facies management are provided in the repository, including signed distances calculations and back transform ("signed_distance_functions.py"), mixed PCA ("dmat_4mixpca.py"). We are working on create a synthetic facies data set so that the user can. We explicitly explained this problem on the provided github AutoBEL repo. we prefer not to write this on the paper because the code will continue to be updated.

• I have some doubts about the back transformation process when not enough PCs are retained. Some artifacts are visible in the realizations on Fig. 19 and on the poro/perm plots of Fig. 20 (breaking the consistency between petrophysical features within each facies). Could this also break the match to borehole data if not enough PCs were selected?

Here, we retain all the PCs of **m** in back transformation. Uncertainty reduction is performed only on the sensitive PCs, while for the non-sensitive PCs of **m**, they remain random according to their prior empirical distribution. But both are used in back transformation. Therefore "not enough PCs" is not a problem in this paper. We further clarified this operation in the methodology section 2.2.2.

• As the aim of the paper is to "minimize the need for tuning parameters" it would be good to summarize the updated parameters as in Table 1 to help the reader assess to what extent the global model parameters have been updated by the process. I cannot help but wondering about how the updating of global parameters such as facies proportions or variogram range would compare to a classical process where statistical inference would be repeated by domain experts as new data become available.

We agree with the referee's suggestion. The updated global parameters in Table 1 are plotted in Figure 11, including the global mean, variogram ranges and facies proportions. In the Figure 11, they are also compared to their prior uncertainty to show the uncertainty reduction from the method. We prefer not to summarize them again in a table as the figure is more effective for the comparison than text table. Note that our method is Bayesian and therefore in accordance with the rules of probability (which domain experts may violate).

• Performance: I assume the 45 minutes do not include the prior model generation? Please clarify.
Yes, the 45 minutes do not include the prior model generation. We have clarified this in the revision.
• Discussion: the discussion in its current form highlights the main points of the method and some challenges, but does not really discuss some aspects which are often considered critical in subsurface models, such as the match of individual realizations to borehole data, or the preservation "geological consistency" such as the
petrophysical distributions for various rock types. I suspect some moderate violations do not really matter for the accumulation problem considered in this paper, but I have more doubts about what would happen for modeling objectives involving highly non-linear physics, such as flow simulation. SOme balanced discussion on these aspects would probably be useful. Another question is whether there are any guidelines about the various sensitivity / confidence levels involved in the method, as these parameters likely impact the results.

We further extended the discussion section to stress more on the critical aspects mentioned by the referee. The extension includes matching individual realizations to local borehole observations and using of BEL for non-linear problems. The discussion section now stands out as a single section. The application of BEL or some BEL steps (e.g. direct forecasting) to non-linear physical problems has been investigated by Satija and Caers (2015), Scheidt et al (2018) on subsurface flows of oil/gas and groundwater reservoirs, and by Hermans et al (2018), Athens and
Caers (2019) on geothermal heat prediction. These works are referred at the introduction to distinguish the unique contributions of our paper.

The guideline about sensitivity study has been thoroughly studied by Fenwick et al (2014) and Park et al (2016). We won't repeat this in the manuscript, but we refereed to these papers in revised section 2.2.1.

p.7, l.11-20: The notations of Eq 11 are unclear to me. I get the point of the sequential updating, but I am not sure it is correct to write "the posterior model of χ becomes prior model for ", as both variables are different. Maybe using the subscript such as χ_posterior in Eq (11) would help make the point clearer.

We rephrased the sentences and revised the equation for more clarity.

P.7, l. 21-25: PCA on an image can be achieved in a variety of manners. I suspect that in Figs. 1 and 2, the PCA factors are linear combinations of image columns, but please explain so that the reader does not have to guess.

Fig 1 and 2 are to show the challenges when performing PCA on categorical models. We added further explanation for this problem.

P.7, l. 27: I'd suggest to use $\psi_s$, as it relates to the distance to facies s. I also think that $\mathbf{x}_\beta$ should be the closest location equal to (and not different from)s in the second term (otherwise, the definition enlarges the facies by 1 voxel). Or maybe just the distance to the boundary of facies $s$?

We changed the notation. $\mathbf{x}_\beta$ is the closest boundary of facies s. We further clarified this.

p.8, l. 7: Typo: the prior uncertainty models *have*

corrected accordingly p.9, l.1-3: Please check convoluted sentence.

Sentence rephased p.9, l.13: Did you mean "the reservoir rocks deposited at shallow marine environments"?

Yes, we added "rocks" to the sentence.

p.10, l.2: Confusion between in-place resources (GIIP) and recoverable reserves, which also depends on flow behavior.

We removed the word "reserve" to only keep GIIP to avoid the confusion.

p.10, l.12-13: Please check syntax.

Syntax corrected

P.10, Fig.5: a scale would be needed so that variogram ranges provided later in the paper can be related to model size and well spacing.

Scale bar is added to the figure.

p.11, table 1: Typo in "gammy ray"

Typo corrected p.12, l.2: In addition to resolution, velocity is a significant source of thickness uncertainty.

We added velocity as another source of uncertainty p.13, l.10-15: h is the height above the free water level, not the reservoir depth.

Corrected according.

p.13, l.19-20: Please fix sentence.

Sentence fixed p.16, l.11: The independence between thickness and facies is stated as a fact. In my view, it is a working hypothesis (probably a reasonable one), but not a general truth.

Yes, the independency between thickness and facies is a fact for the case, not a general truth. We rephrased the sentence to gain clarity.

p. 18, Fig.10: Would dobs correspond to a line? I guess it should have some thickness due to data noise and to PCA projection.

We agree with the referee on this. For this synthetic case, we didn't consider the data noise, so it is one value represented by a line.

p.19, l.1-2: "the uncertainty... their prior": Please chech grammar.

Grammar checked and corrected for this line.

p.19, Fig.11: Please tell what the dash lines correspond to (kernel density?)

The dash lines are the estimated probability density using Gaussian kernels. We add this explanation to the figure caption.

p.20, Fig. 12: The visualization for posterior facies distributions gives a qualitative hint about what happens, but I am not sure about what we are exactly looking at. It is the blending of discrete color maps or the average of the underlying Gaussian field (but then, facies threshold change depending on facies proportions)?

We agree with the referee's comments on the Figure 12. This figure has been replaced by the median prior and posterior model. This is also suggested by Referee #1.

Fig. 15: Typo in legend

Typo corrected p.25: Please remove mention to reserves, as no recovery factor is involved.

"reserves" removed in the revision p.25, l.17: The cross-validation tests on wells 5 and 6 seems a bit optimistic, as vertical averaging on the 75 layers essentially amounts to making a two-dimensional model. Again, erors average out, so the reduction of uncertainty in such a case is no proof of the actual forecasting ability of the model in three dimensions.

The aim here is to test the resulting posterior models are not wrong and can predict future new borehole observation with reduced uncertainty. We rephased these lines near l.17 by removing the statements on forecasting ability to avoid the ambiguity and stress on our main point.

p. 24, Fig. 20: The correlation coefficient is not really meaningful on such multi-modal distributions, even more so as the facies proportions likely change from one distribution to another. It would make more sense to examine the bivariate statistics per facies.

We have replotted the Fig 20 to examine the bivariate statistics facies by facies.

P.26, Fig. 23: please explain what we see: the curves are densities, but what does the point horizontal spread mean? And again, what is the facies value?

The horizontal spread of the scatters is for better visualization. Otherwise, the points will overlap each other, making it difficult to observe their distributions. The facies values on the plot are averaged across the 75 layers.

p.27 l.12, l.15, l.24 : "results an extreme fast computation"; "be able predict", "do not full match": Please fix English

These sentences are fixed.

p.28, L.11-21: This paragraph nicely explains the problem in simple terms, so I think it would be better integrated in the introduction than in the discussion.

We put this paragraph to the newly created discussion chapter because it fits better to the context below and above.

p.28, L.16: this sentence mixes the falsification of parameters and falsification of a methodology, which I think are very different. I suggest rephrasing.

The sentence is rephrased to focus on the problem from falsified prior model.

p.28, l.30: Not sure I understand the references in this context.

We rephrased the sentence.

**Responses to Tao Feng (taof76@hotmail.com) on interactive comments (SC1)**

Dear authors,

This paper presents an automated workflow to build geo-model using some hard data. The work is very interesting, and the paper is well written. I strongly suggest publication after some revision. I have a few questions and comments. I hope authors can clarify it.

We thank the reviewer for the interest and comments on our paper. Please see below our responses.

1. In BEL, you mentioned prediction. There is nothing related prediction from the field case, am I right? It seems to me this work is mainly related model building using data. The prediction variable h is the model parameters, right?

In the field case, the prediction variable **h** is reservoir storage volume, which is directly calculated from the models. Our main work was to develop an automated approach (Auto-BEL) to jointly update spatial models and storage volume uncertainties using data. The steps of Auto-BEL include build/update models using data.

2. Using observation d_obs, you can detect some outlier realizations. Do you just remove these realizations from your prior in practice?

No, we didn't remove these realizations. We keep all the prior realizations if they are not falsified. Removing the outlier realizations can change the prior distribution.

3. h*, d* are some subspace of h and d, right? If we talking permeability field with millions of cells, could you give me roughly number of h* compared to h? Is Formula (9) standard way to formula linear-Gauss problem?

Yes, h*, d* are subset of orthogonalized h and d (after PCA, and CCA). The dimension of h* is the total number of model realizations, although h can be up to millions of cells. Yes, for linear-Gauss problem, Formula (9) is standard.

4. Your Python tool can be used to build geo-model (grid, etc..)?

Yes, it can directly update/rebuild geo-models, once new borehole data are provided, without using any external modelling tools. But it requires Monte Carlo of prior geo-models for "training". These prior models were built with a geomodeling tool (Petrel)

5. After CCA, elements in h* is more independent (less correlated), right?

We don't think so. CCA didn't change the independency of h* elements. h* elements are independent already before CCA, because they are orthogonalized via PCA.

6. I am very interested in "sequential update model". It will be nice if the authors can describe this in more details.

Thanks for the suggestion. We have added a paragraph to the application section (section 3.4.2) to explicitly explain how the sequential approach is performed.

7. Every time you update a parameter, do you use the posterior as a prior for next parameter update?

Exactly, the posterior from previous sequence is used as prior for the next update.

**Responses to Dexiang Li (dexiangli_pe@163.com) on interactive comments (SC2)**

Geological models can be more accurate and actual with coupling more borehole data in models. Meanwhile, the
data sizes of geological models increase with the developments of field projects and participation of new borehole data. During dynamic process of subsurface applications such as groundwater, geothermal, oil, gas, and CO2 geostorage, uncertainty quantification is the key for decision making. As the authors mentioned, uncertainty reduction is a time-consuming work which requires iterative model rebuilding using conventional inverse methods. In order to make the model adhere to geological rules, geological modeling often requires significant
individual/group ex-pertise and manual intervention which will need often months of work after new data is achieved. In this paper, the authors generalized a Monte Carlo-based framework for geological uncertainty quantification and updating. Their methodologies were developed with the BEL protocol for uncertainty quantification. The extension of directly forecasting results an extreme fast computation of posterior geological model, by avoiding conventional model rebuilding. The proposed framework also allows automation of geological
UQ. This paper is interesting and in an area worthy of investigation. Overall, this paper is well-organized and well-written. This paper can be accepted by addressing the following minor comments.

We appreciate the independent reviewer's in-depth understanding of our paper. The comments are helpful to improve our paper. Please see below our responses and explanations on the revision.

1. The advantages and disadvantages using your method for UQ and updating should be further illustrated by
comparing with typical conventional method. At the same time, its applicable scenarios are suggested to be provided which can give guidelines for field application.

We extended the discussions on limitations in the last paragraph of "Discussion and conclusions". The advantages of this method have been discussed in second and third paragraph (page 27, 28). The application scenarios are for uncertainty quantification using borehole data in geological modeling and prediction, which are common in
oil&gas, geothermal, CO2 sequestration applications. We have added a statement at the beginning of abstract for more context of this.

2. As you mentioned, current method is only designed to globally adjust the model, not locally at the borehole observation. Could you provide your idea on further solution in more details?
As we explained at the discussion section, one possible solution we like to explore is to combine geostatistical conditional simulation as posterior step to our current methodology. For example, once the posterior global parameters are calculated from AutoBEL, they can be used as the input to geostatistical simulation conditioned to the local well observations. This will enable posterior models locally matched to the borehole observations with reduced global uncertainty.

3. Could provide the specific performance parameters of CPU which can show the improvement on calculation efficiency more accurately?

The CPU is Intel Core i7-7820HQ. We added this specification to the revision.

4. The authors are suggested to unify the multiplication sign through the whole manuscript?

We unified the multiplication symbols in the revision.

5. Please add a "." between "Figure 19" and the "Prior and posterior..." to keep in accordance with other figures. Please check similar problems accordingly.

"." has been added to the figure captions.

6. The usage of abbreviation such as DF should be noticed. Abbreviations should be defined when they are first mentioned in the text and should always be used afterwards.

The abbreviation DF is defined at Page6/Line4

7. Discussion and conclusions are suggested to be separated into two parts. Please provide conclusions point by point which can help reader to understand the main con-tributions of the paper. Meanwhile, future researches should be clarified according to the limitations of proposed method.

Thanks for this suggestion. Discussion section now stands as a single section for more in-depth discussions. The conclusions are point by point already. In paragraph 1 of this section, we have clearly itemized the contributions by words such as "generalized", "second contribution", "third contribution" … We extended the discussion on future researches and provided relevant references in the new revision.

---

## Author Comment (AC3) · 10 Jan 2020

Dear Guillaume Caumon,

Thanks for taking time to review our paper "Automated Monte Carlo-based Quantification and Updating of Geological Uncertainty with Borehole Data (AutoBEL v1.0)". We highly appreciate your comments and suggestions. Please find blew our responses which are highlighted in red.

On behalf of the Authors,
Zhen Yin (David)

**Responses to Guillaume Caumon (Referee) on interactive comments (RC2)**

**Guillaume Caumon (Referee)**

guillaume.caumon@ensg.univ-lorraine.fr

**General comments**

This paper proposes an application of the "Bayesian Evidential Learning" approach to the problem of uncertainty assessment in integrated reservoir modeling. Although the general approach was previously described in Scheidt et al (2018), this paper contains significant new elements, applications and discussions, which are very interesting for the community.

In terms of form, the paper is very well written and clearly presented, apart from minor issues. It includes a link to a Jupyter notebook implementing the methodology and applying it to the reservoir thickness. The implementation works fine, after some twiddling to install scikit-fmm. The overall structure of the code seems to allow for managing facies (some unused functions for facies modeling are present in the code), but the demo notebook assumes porosity to be 1 and water saturation to be zero. Even so, the reproducibility is much better than in most similar papers on this topic.

Overall, I congratulate the authors for the very interesting approach which represents a paradigm shift as compared to the current practice. I have, nonetheless, several comments, questions, and suggestions, which I hope will help to improve the paper. My recommendation is to proceed with minor/moderate revision.

We are very thankful to the referee's thorough and in-depth review. The comments almost cover all the aspects of our paper. They are insightful and very helpful to improve our work. We overall agree with them. Provided below shows how the referee's comments are addressed point-to-point and incorporated to the revised manuscript

**Specific comments**

• Overall, the introduction makes a good job introducing the general problem, but more precise explanations about the exact contributions of the paper would be welcome at this stage (in particular with regard to the other recent contributions of SCREF).

We rephrased the last paragraph of "Introduction" to clarify the exact contributions of our paper with regard to the previous papers on this topic (of BEL).

The main contributions are

- To propose a model falsification scheme using robust Mahalanobis distance.

- Extension direct forecasting based on sequential model decomposition to honor the hierarchical rules in geological modeling.
- A complete automation of geological uncertainty quantification using borehole data.

5 • The borehole data are generally at much higher resolution than the reservoir grid data. However, as in most reservoir modeling approaches, this paper assumes that the borehole data has been upscaled to grid resolution, a process which is source of inaccuracies in reservoir models. One of the points of the proposed approach is that the falsification step (Section 2.1.3) could in principle integrate the scale change. Comments on this would be welcome. Also, some additional precisions about the management of categorical variables during falsification

10 would be welcome (in addition to the last sentence on page 5). From Eq. (8), it seems that the robust Mahalanobis distance accounts for spatial redundancy; please confirm.

In this paper we deal with borehole observations that are already upscaled to model grid resolution. Upscaling errors is not within the scope of this paper's research, but is for us a very active area of our current research. The way we approaching this problem is to add such error to the data covariance matrix in Eq10, using some Monte

15 Carlo analysis, but as said, see our next paper. As a result the falsification then indeed tests only a necessary condition (upscaled data) but not a sufficient test (actual data)

One rather ad-hoc trick around that is to choose a lower threshold value as the tolerance of Chi-squared distribution of $\mathrm{RMD}(\mathbf{d})$ (making the test more powerful) We have added a few sentences in Section 2.1.3 to

20 further clarify this problem.

Yes, spatial redundancy is accounted by performing dimension reduction (PCA) on the well data. Eq (8) uses the PC scores of $\mathbf{d}$.

25 • Overall, I am not fully clear about the falsification step. As this is a key aspect of the proposed methodology, it would be good if the authors could insist on the consequences of this approach as compared to the conventional method which creates models sampling exactly the borehole data. Errors between model and data may be acceptable for some applications such as hydrocarbon in place, as they will average out, but would they yield reliable forecasts of porous flow and transport if borehole data are not exactly honored by some model

30 realizations?

The goal of falsification is not to check errors between model and data, but to check whether the model can predict the data, even if we would account for some tolerance (or error). If that is not the case, then one would be averaging out over a wrong model. Therefore, we are not looking here at variance of error, but bias in the prior

35

• Although this is not the main point of the paper, some parameters for generating the prior models could be described more precisely and discussed. For example, I am a bit puzzled by uniform distributions taken for the facies 1,2,3, which suggests that facies 0 will adjust so that the total is equal to 1, which may be a source of bias (see Haas Formery, Math. Geol. 2002, or compositional data analysis literature). What are the variogram ranges for facies modeling? In the figure, there seems to be a facies trend, but what are the parameters of this trend? Are the variogram models isotropic? Is the variogram of porosity the same for all facies?

Yes, the prior modeling is not the focus of our paper. We therefore prefer not to stress much on the prior modeling, but to focus more on our main contributions on falsification, direct forecasting on sequential decomposition and automation. Additionally, Referee #1 suggests to remove specific explanations on the prior modeling to gain more generality (see General comment #2 of Referee #1). We agree with this comment.

Regarding the prior facies modeling, a deterministic trend was applied. This is to make sure that the orientation and position of modeled facies belts are consistent with geological interpretations in Figure 5(a). The trend in the case study was provided by the field geologists, and officially used by the field operator. The variogram models are isotropic for facies, anisotropic for porosity. The variogram of porosity is the same in all facies. We quantify the prior model uncertainty under this scenario.

• Overall, I get the overall idea for facies, but I am not fully clear about the consequence of the facies processing. The signed distance is mentioned, and then the Truncated Gaussian. I first understood that the TG was used in the generation of prior models, and the signed distance for the workflow steps (which would mean in general 3 distance fields for 4 facies). But I was then puzzled by Fig 12 which suggests that maybe one single scalar field is used. So at the end, I am not sure about what was done exactly. Clarification of this would be needed in the paper.

The referee is correct on understanding the overall idea for facies uncertainty quantification. We agree that the main problem is from Fig 12 which is confusing. We replotted Fig 12 to avoid such confusions. The figure is replaced by the median posterior facies model to show the final results from Auto-BEL, which is also suggested by Referee #1.

• I looked at the code to try to understand the facies management, but it is not fully integrated in the high-level functions. Adding facies management in the code would improve reproducibility. If not possible, then please explicitly mention in Section 2.3 that the provided python code illustrates the workflow for thickness only.

We appreciate the referee's test of the Auto-BEL code. Regarding the problem of facies management, we couldn't provide the facies model data because it is classified as confidential by the company. However, all the code functions for facies management are provided in the repository, including signed distances calculations and back transform ("signed_distance_functions.py"), mixed PCA ("dmat_4mixpca.py"). We are working on create a synthetic facies data set so that the user can. We explicitly explained this problem on the provided github AutoBEL repo. we prefer not to write this on the paper because the code will continue to be updated.

• I have some doubts about the back transformation process when not enough PCs are retained. Some artifacts are visible in the realizations on Fig. 19 and on the poro/perm plots of Fig. 20 (breaking the consistency between petrophysical features within each facies). Could this also break the match to borehole data if not enough PCs were selected?

5 Here, we retain all the PCs of **m** in back transformation. Uncertainty reduction is performed only on the sensitive PCs, while for the non-sensitive PCs of **m**, they remain random according to their prior empirical distribution. But both are used in back transformation. Therefore "not enough PCs" is not a problem in this paper. We further clarified this operation in the methodology section 2.2.2.

10 • As the aim of the paper is to "minimize the need for tuning parameters" it would be good to summarize the updated parameters as in Table 1 to help the reader assess to what extent the global model parameters have been updated by the process. I cannot help but wondering about how the updating of global parameters such as facies proportions or variogram range would compare to a classical process where statistical inference would be repeated by domain experts as new data become available.

15 We agree with the referee's suggestion. The updated global parameters in Table 1 are plotted in Figure 11, including the global mean, variogram ranges and facies proportions. In the Figure 11, they are also compared to their prior uncertainty to show the uncertainty reduction from the method. We prefer not to summarize them again in a table as the figure is more effective for the comparison than text table. Note that our method is Bayesian and therefore in accordance with the rules of probability (which domain experts may violate).

20 • Performance: I assume the 45 minutes do not include the prior model generation? Please clarify.

Yes, the 45 minutes do not include the prior model generation. We have clarified this in the revision.

• Discussion: the discussion in its current form highlights the main points of the method and some challenges, but does not really discuss some aspects which are often considered critical in subsurface models, such as the match of individual realizations to borehole data, or the preservation "geological consistency" such as the

25 petrophysical distributions for various rock types. I suspect some moderate violations do not really matter for the accumulation problem considered in this paper, but I have more doubts about what would happen for modeling objectives involving highly non-linear physics, such as flow simulation. SOme balanced discussion on these aspects would probably be useful. Another question is whether there are any guidelines about the various sensitivity / confidence levels involved in the method, as these parameters likely impact the results.

30 We further extended the discussion section to stress more on the critical aspects mentioned by the referee. The extension includes matching individual realizations to local borehole observations and using of BEL for non-linear problems. The discussion section now stands out as a single section. The application of BEL or some BEL steps (e.g. direct forecasting) to non-linear physical problems has been investigated by Satija and Caers (2015), Scheidt et al (2018) on subsurface flows of oil/gas and groundwater reservoirs, and by Hermans et al (2018), Athens and

35 Caers (2019) on geothermal heat prediction. These works are referred at the introduction to distinguish the unique contributions of our paper.

The guideline about sensitivity study has been thoroughly studied by Fenwick et al (2014) and Park et al (2016). We won't repeat this in the manuscript, but we refereed to these papers in revised section 2.2.1.

References:

Fenwick, D., Scheidt, C. and Caers, J.: Quantifying Asymmetric Parameter Interactions in Sensitivity Analysis: Application to Reservoir Modeling, Math. Geosci., 46(4), 493–511, doi:10.1007/s11004-014-9530-5, 2014.

Park, J., Yang, G., Satija, A., Scheidt, C. and Caers, J.: DGSA: A Matlab toolbox for distance-based generalized sensitivity analysis of geoscientific computer experiments, Comput. Geosci., 97, 15–29, doi:10.1016/J.CAGEO.2016.08.021, 2016.

Satija, A. and Caers, J.: Direct forecasting of subsurface flow response from non-linear dynamic data by linear least-squares in canonical functional principal component space, Adv. Water Resour., 77, 69–81, doi:10.1016/J.ADVWATRES.2015.01.002, 2015.

Scheidt, C. Ã., Li, L. and Caers, J.: Quantifying Uncertainty in Subsurface Systems, Wiley. [online] Available from: https://books.google.com/books?id=xvRYDwAAQBAJ, 2018.

Athens, N. D. and Caers, J. K.: A Monte Carlo-based framework for assessing the value of information and development risk in geothermal exploration, Appl. Energy, 256, 113932, doi:10.1016/J.APENERGY.2019.113932, 2019a.

**Technical corrections**

In several places: the term "data-scientific" looks hype but I don't get the exact meaning. lease define or use another term.

We replaced the term "data-scientific" by "statistical learning" for more precise description of our approach.

Also, the term "constraint" used in several places could probably be replaced with more accurate terminology.

We rephrased the sentences to remove the term "constraint" to for more accuracy.

p.1, l.19: "A generalized synthetic data set motivated by a gas reservoir": please rephrase. Seems to me this is a gas reservoir study which has been simplified.

The statement is rephrased to "a generic gas reservoir dataset.".

Yes, this is a reservoir simplified from real gas reservoir for general research implementations.

p.4, Eq. (3): Unless I am missing something, the notation could be simpler using $\chi_{gl}, \chi_{sp}$ etc.

Notation corrected accordingly

p.6, l.10: Please define G (and make it bold?). Fix typos in Eq. (10): "proir" should read prior.

**G** are bolded and defined in the revised Eq (10). Typos corrected accordingly.

p.6, l.24-25: Please rephrase the sentence for more clarity. This is more an explanation about why it works in practice than an actual "truth".

We remove this sentence because it introduces confusion, and previous explanations has already explained the problem.

p.7, l.1: Instead of "model grid cells", I'd suggest for generality: model parameters

Revised accordingly

p.7, l.3-6: I'd recommend to factually summarize the DGSA approach rather than summarizing its advantages (as compared to what?).

We rephrased the sentences to compare DGSA to the other global sensitivity analysis methods such as variance-based methods (e.g. Sobol, 2001, 1993), regionalized methods (e.g. Pappenberger et al., 2008; Spear and Hornberger, 1980), or tree-based method (e.g. Wei et al., 2015).

References:

Sobol, I. .: Global sensitivity indices for nonlinear mathematical models and their Monte Carlo estimates, Math. Comput. Simul., 55(1–3), 271–280, doi:10.1016/S0378-4754(00)00270-6, 2001.

Sobol, I. M.: Sensitivity estimates for nonlinear mathematical models., Math. Model. Comput. Exp., 1(4), 407–414, 1993.

Spear, R. C. and Hornberger, G. M.: Eutrophication in peel inlet—II. Identification of critical uncertainties via generalized sensitivity analysis, Water Res., 14(1), 43–49, doi:10.1016/0043-1354(80)90040-8, 1980.

Wei, P., Lu, Z. and Song, J.: Variable importance analysis: A comprehensive review, Reliab. Eng. Syst. Saf., 142, 399–432, doi:10.1016/J.RESS.2015.05.018, 2015.

p.7, l.11-20: The notations of Eq 11 are unclear to me. I get the point of the sequential updating, but I am not sure it is correct to write "the posterior model of $\chi$ becomes prior model for ", as both variables are different. Maybe using the subscript such as $\chi\_posterior$ in Eq (11) would help make the point clearer.

We rephrased the sentences and revised the equation for more clarity.

P.7, l. 21-25: PCA on an image can be achieved in a variety of manners. I suspect that in Figs. 1 and 2, the PCA factors are linear combinations of image columns, but please explain so that the reader does not have to guess.

Fig 1 and 2 are to show the challenges when performing PCA on categorical models. We added further explanation for this problem.

P.7, l. 27: I'd suggest to use $\psi_s$, as it relates to the distance to facies s. I also think that $\mathbf{x}_\beta$ should be the closest location equal to (and not different from)s in the second term (otherwise, the definition enlarges the facies by 1 voxel). Or maybe just the distance to the boundary of facies $s$?

We changed the notation. $\mathbf{x}_\beta$ is the closest boundary of facies s. We further clarified this.

p.8, l. 7: Typo: the prior uncertainty models *have*

corrected accordingly

p.9, l.1-3: Please check convoluted sentence.

Sentence rephased

p.9, l.13: Did you mean "the reservoir rocks deposited at shallow marine environments"?

Yes, we added "rocks" to the sentence.

p.10, l.2: Confusion between in-place resources (GIIP) and recoverable reserves, which also depends on flow behavior.

We removed the word "reserve" to only keep GIIP to avoid the confusion.

p.10, l.12-13: Please check syntax.

Syntax corrected

P.10, Fig.5: a scale would be needed so that variogram ranges provided later in the paper can be related to model size and well spacing.

Scale bar is added to the figure.

p.11, table 1: Typo in "gammy ray"

Typo corrected

p.12, l.2: In addition to resolution, velocity is a significant source of thickness uncertainty.

We added velocity as another source of uncertainty

p.13, l.10-15: h is the height above the free water level, not the reservoir depth.

Corrected according.

p.13, l.19-20: Please fix sentence.

Sentence fixed

p.16, l.11: The independence between thickness and facies is stated as a fact. In my view, it is a working hypothesis (probably a reasonable one), but not a general truth.

Yes, the independency between thickness and facies is a fact for the case, not a general truth. We rephrased the sentence to gain clarity.

p. 18, Fig.10: Would dobs correspond to a line? I guess it should have some thickness due to data noise and to PCA projection.

We agree with the referee on this. For this synthetic case, we didn't consider the data noise, so it is one value represented by a line.

p.19, l.1-2: "the uncertainty... their prior": Please chech grammar.

Grammar checked and corrected for this line.

p.19, Fig.11: Please tell what the dash lines correspond to (kernel density?)

The dash lines are the estimated probability density using Gaussian kernels. We add this explanation to the figure caption.

p.20, Fig. 12: The visualization for posterior facies distributions gives a qualitative hint about what happens, but I am not sure about what we are exactly looking at. It is the blending of discrete color maps or the average of the underlying Gaussian field (but then, facies threshold change depending on facies proportions)?

We agree with the referee's comments on the Figure 12. This figure has been replaced by the median prior and posterior model. This is also suggested by Referee #1.

Fig. 15: Typo in legend

Typo corrected

p.25: Please remove mention to reserves, as no recovery factor is involved.

"reserves" removed in the revision

p.25, l.17: The cross-validation tests on wells 5 and 6 seems a bit optimistic, as vertical averaging on the 75 layers essentially amounts to making a two-dimensional model. Again, erors average out, so the reduction of uncertainty in such a case is no proof of the actual forecasting ability of the model in three dimensions.

The aim here is to test the resulting posterior models are not wrong and can predict future new borehole observation with reduced uncertainty. We rephased these lines near l.17 by removing the statements on forecasting ability to avoid the ambiguity and stress on our main point.

p. 24, Fig. 20: The correlation coefficient is not really meaningful on such multi-modal distributions, even more so as the facies proportions likely change from one distribution to another. It would make more sense to examine the bivariate statistics per facies.

We have replotted the Fig 20 to examine the bivariate statistics facies by facies.

P.26, Fig. 23: please explain what we see: the curves are densities, but what does the point horizontal spread mean? And again, what is the facies value?

The horizontal spread of the scatters is for better visualization. Otherwise, the points will overlap each other, making it difficult to observe their distributions. The facies values on the plot are averaged across the 75 layers.

p.27 l.12, l.15, l.24 : "results an extreme fast computation"; "be able predict", "do not full match": Please fix English

These sentences are fixed.

p.28, L.11-21: This paragraph nicely explains the problem in simple terms, so I think it would be better integrated in the introduction than in the discussion.

We put this paragraph to the newly created discussion chapter because it fits better to the context below and above.

p.28, L.16: this sentence mixes the falsification of parameters and falsification of a methodology, which I think are very different. I suggest rephrasing.

The sentence is rephrased to focus on the problem from falsified prior model.

p.28, l.30: Not sure I understand the references in this context.

We rephrased the sentence.